# AIDA Arctic transport experiment (part 1): simulation of northward transport and aging effect on fundamental black carbon properties

Zanatta, Marco[1,2], Bogert, Pia[1], Ginot, Patrick.[3], Gong, Yiwei.[1,a], Hoshyaripour, Gholam Ali[4], Hu, Yaqiong.[1], Jiang, Feng.[1,b], Laj, Paolo[3,c], Li, Yanxia.[1], Linke, Caudia[1], Möhler, Ottmar.[1], Saathoff, Harald.[1], Schnaiter, Martin.[1,d], Umo, Nsikanabasi Silas[1], Vogel, Franziska.[1,e], Wagner, Robert[1]

[1] Institute of Meteorology and Climate Research – Atmospheric Aerosol Research, Karlsruhe Institute of Technology, Karlsruhe, Germany
[2] Institute of Atmospheric Science and Climate, National Research Council of Italy, Bologna, Italy
[3] University Grenoble Alpes, CNRS, IRD, G-INP, Institut des Géosciences de l'Environnement, Grenoble, France
[4] Institute of Meteorology and Climate Research Troposphere Research, Karlsruhe Institute of Technology, Karlsruhe, Germany
[a]Now at: PSI Center for Energy and Environmental Sciences, 5232 Villigen PSI, Switzerland
[b]Now at: School of Public and Environmental Affairs, Indiana University, Bloomington IN 47401-2204,USA
[c]Now at: World Meteorological Organization, Geneva, Switzerland
[d]Now at: Institute for Atmospheric Research of Wuppertal, Wuppertal, Germany
[e]Now at: Institute of Atmospheric Science and Climate, National Research Council of Italy, Bologna, Italy

*Correspondence to: m.zanatta@isac.cnr.it*
*Correspondence to: gholamali.hoshyaripour@kit.edu*

**Abstract** Black carbon (BC) is a key atmospheric forcer due to its interaction with solar radiation and clouds. However, accurately quantifying and understanding the impact of atmospheric aging on BC properties and radiative forcing remains a major challenge. To address this, the AIDA aRCtic Transport Experiment (ARCTEx) project simulated BC aging under quasi-realistic Arctic conditions in the AIDA (Atmospheric Interactions and Dynamics in the Atmosphere) chamber. Four distinct scenarios were simulated based on reanalysis data, representing summer and winter conditions at both low and high altitudes, to capture the variability in BC aging processes in the presence of nitrate and organic matter precursors during Arctic transport.

In the first part of the paper, we define the meteorological conditions characterizing northward transport under different scenarios and describe the technical solutions to simulate 5-day transport in the AIDA chamber. In the second part of the work, we assess the evolution of fundamental properties including density, morphology and mixing state observed during the aging process.

The ARCTEx project demonstrates that large facilities such as AIDA can successfully reproduce environmental conditions, enabling a gradual aging process that closely follows the natural timescales observed in the atmosphere. Our experiments revealed that temperature strongly influences the aging timescale and the evolution of BC's diameter, effective density, and coating thickness. Low-altitude scenarios exhibited rapid aging, resulting in fully-coated, compact BC particles within 39 – 98 hours, corresponding to 50°N and 80°N respectively. In contrast, high-altitude transport was characterized by slow aging, with limited coating and compaction, even after 115 hours of simulation. These findings provide valuable insights into the temporal evolution of BC properties during Arctic transport. In forthcoming work, we will report the implications of this evolution on climate-relevant properties such as light absorption and activation as cloud droplets and ice crystals. Together, these studies aim to enhance the representation of BC aging in climate models, reducing uncertainties in Arctic radiative forcing estimates.

## 1 Introduction

Black carbon (BC) is a primary carbonaceous aerosol emitted by combustion processes. Due to its strong absorption of visible light, BC is the only aerosol exerting a net warming effect (IPCC, 2023). As

described by Schulz et al. (2006), the direct radiative forcing of BC is proportional to its mass absorption cross-section (MAC) and its atmospheric lifetime. While MAC quantifies the amount of light absorbed per unit mass of BC, the lifetime depends on BC's ability to act as cloud condensation nuclei (CCN) and ice-nucleating particles (INPs). These three climate-relevant properties (MAC, CCN, and INP) are directly influenced by BC's fundamental physical and chemical characteristics, including particle

diameter, mixing state, and morphology (Bond et al., 2013). However, these properties vary during the lifetime of BC due to atmospheric aging. Consequently, also the radiative forcing of BC varies as a function of its atmospheric age. As summarized by Li et al. (2024), mixing with other chemical substances can enhance BC's net light absorption and increase its hygroscopicity. However, significant uncertainties remain: i) the magnitude of absorption enhancement is still debated (Cappa et al., 2012), ii) closure studies

are not yet robust to confirm BC's role as a CCN (Bond et al., 2013), iii) BC's freezing efficiency in cirrus clouds is still unclear (Burrows et al., 2022).

Considering the long lifetime of BC in the Arctic (5.5 days, Lund et al., 2018), the impact of aging on the evolution of BC's fundamental and climate-relevant properties during Arctic long-range transport remains challenging to be reproduced in global models (Lund and Berntsen, 2012; Mahmood et al., 2016).

This generates significant uncertainties in the estimation of BC concentrations and radiative forcing in the Arctic region (Samset et al., 2013). Moreover, the limited observational capability of aging-induced modification of BC does not allow fully validating the performances of global models (Samset et al., 2018). In fact, most measurements in the Arctic provide only static snapshots of BC properties, offering limited insight into the dynamic processes occurring during the transport and leading to internal mixing

(Kodros et al., 2018), absorption enhancement (Zanatta et al., 2018), and cloud activation (Zanatta et al., 2023; Zieger et al., 2023) in the Arctic region.

On the other hand, experiments conducted in laboratory set-ups using flow tubes and simulation chambers have been essential in advancing our understanding of the aging processes of aerosol and black carbon particles (Doussin et al., 2023). In general, these studies reported a significant modification of BC

properties as a function of chemical and physical aging, including morphology (e.g. Saathoff et al., 2003a; Corbin et al., 2023), light absorption (e.g. Schnaiter et al., 2005; Fierce et al., 2020), hygroscopicity (e.g. Henning et al., 2012; Dalirian et al., 2018; Friebel and Mensah, 2019) and ice nucleation activity (e.g. Möhler et al., 2003a; Kanji et al., 2020). Although chamber experiments on BC aging have been actively implemented to constrain global models (Wang et al., 2018), they are often conducted under controlled

conditions that may not fully represent real-world atmospheric environments. These experiments typically involve high concentrations, short time scales and limited variability in temperature, humidity, and chemical composition. As a result, aging processes may occur with different timescales and effects compared to ambient atmospheric conditions, introducing challenges in translating experimental findings into model parameterizations. Improving the representation of BC aging in models is crucial for reducing

uncertainties in climate projections, as accurate simulations of BC properties and evolution are essential for assessing its radiative forcing and cloud interactions (Wang et al., 2018).

BC variability in the Arctic was often associated with co-emitted sulfate aerosol from anthropogenic sources (e.g. Massling et al., 2015) and organic aerosol from biomass burning events (e.g. Moschos et al., 2022), while its correlation with nitrate was mostly ignored (AMAP, 2021). Similarly, chamber studies

focused on the evolution of BC properties as function of internal mixing with sulfate (e.g. Möhler et al., 2005; Khalizov et al., 2009; Henning et al., 2012) and organics (e.g. Lefevre, 2019; Wittbom et al., 2014). As a consequence, the impact of BC-nitrate internal mixing on fundamental and climate relevant properties remained poorly assessed (Yuan et al., 2020). Internal mixing of BC with nitrate species becomes particularly important in the Arctic region where nitrate aerosol concentration has been

increasing since the '80s despite an overall reduction of nitrogen oxides emissions (AMAP, 2021). The same report underlined how few studies had focused on nitrate aerosol in the Arctic, introducing a knowledge gap on the sources of its precursors, its formation mechanisms and its interaction with other atmospheric species such as BC.

ARCTEx (AIDA aRCtic Transport Experiment) was designed to reproduce in the AIDA (Aerosol

Interaction and Dynamics in the Atmosphere) chamber the meteorological and chemical conditions that BC undergoes during Arctic transport, focusing on mixing with nitrogen-based species. By simulating different transport scenarios, we aim to understand and quantify the aging timescale and the impacts on fundamental and climatic-relevant properties of BC during different seasons and altitudes. In the present

work, we present the proof of concept of ARCTEx, including the definition of Arctic transport conditions, a full technical description of the unprecedented 5 day-long experiments and the evolution of fundamental properties such as diameter, density, morphology, and mixing state. A second publication will aim to explore the relationship of the aging timescale explored here with climate-relevant properties.

## 2 Methods

The objective of the ARCTEx is to quantify the impact of aging on BC's fundamental and climate-relevant properties during long-range transport from mid-latitudes to the Arctic. This is achieved through chamber simulation experiments. The methodology section is structured as follows: first, we describe the identification of Arctic transport scenarios (Section 2.1) to be reproduced in the Aerosol Interaction and Dynamics in the Atmosphere (AIDA) chamber (Section 2.2). The procedures for injecting and measuring reactive gases and aerosol particles are detailed in Section 2.3 and Section 2.4, respectively. Finally, an overview of the experimental design and schedule is provided in Section 2.5.

### 2.1 Identification of Arctic transport scenarios

#### 2.1.1 Region of interest

Aerosol particles can be transported from mid-latitudes to the Arctic region following various paths. As summarized in the AMAP report 2015 (Quinn et al., 2015), the transport of pollution is regulated by the seasonal variability of the "Arctic front" (Barrie, 1986), which controls the geographical origin of pollution and the altitude of the transport pathway. The Eurasian sector is a significant corridor for northward transport, and it is of particular interest to the member and observer countries of the Arctic Council due to very high BC emissions (Schacht et al., 2019). This sector is associated with intense export of anthropogenic (Backman et al., 2021) and natural (McCarty et al., 2021) black carbon emissions. Although other efficient transport patterns exist, the continental Eurasian sector is characterized by a reduced temperature variability (Przybylak, 2016), ensuring more homogeneous atmospheric conditions compared to the Atlantic and Pacific transport pathways. This uniformity is critical for the design of chamber experiments, as it minimizes external variability and facilitates the representation of the ambient conditions, thereby reducing the operational complexity of the AIDA-chamber. Hence, we identified the area of interest for ARCTEx in the Eurasian sector comprised between 60°E and 140°E longitude and between 40°N and 90°N latitude (Figure 1).

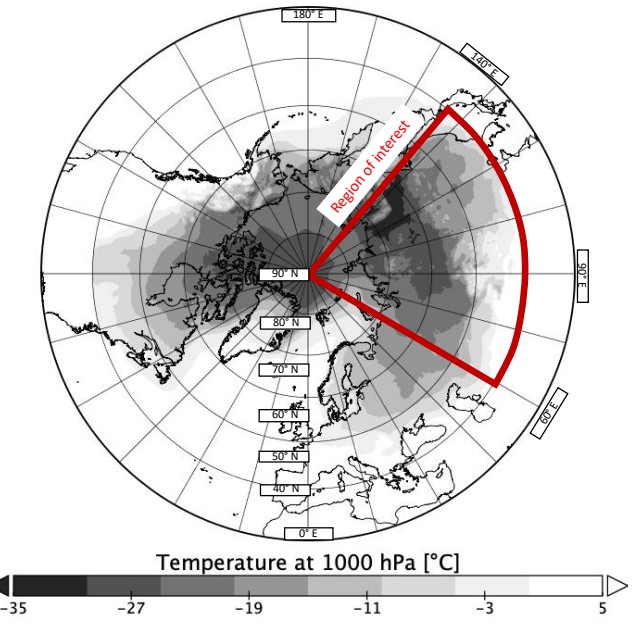

**Figure 1. Global variability of atmospheric temperature at 1000 hPa as extracted from ERA5 monthly averaged data (mean of the year 2010). Marked in red is the region of interest for the ARCTEx project, limited between 40-90°N and 60-140°E.**

### 2.1.2 Transport pathways

In winter and early spring, the southern extent of the polar front (around 40°N) enables the rapid transport of pollution emitted from mid-latitudes in the Eurasian and North American regions to the lower troposphere of the Arctic. In summer, warm air masses from south of the Arctic front can move northward, maintaining a constant potential temperature. As they rise, these air masses reach the Arctic in the mid- and upper troposphere, but over a longer timescale compared to the faster winter transport. This seasonal variability results in a clear seasonality of BC concentration, with higher values in the winter-spring and lower ones in summer across the low and mid troposphere (Jurányi et al., 2023). In this work, January and July were assumed to be representative of the winter and summer season. In terms of altitude, we assumed the 1000 - 800 hPa of atmospheric pressure as representative of low-altitude (low troposphere) and 600 - 400 hPa of atmospheric pressure as representative of high-altitude (mid troposphere).

Overall, we recognized four distinct transport scenarios:

- Summer (July) low-altitude (1000 - 800 hPa) transport (SL)

- Winter (January) low-altitude (1000 - 800 hPa) transport (WL)

- Summer (July) high-altitude (600 - 400 hPa) transport (SH)

- Winter (January) high-altitude (600 - 400 hPa) transport (WH)

For these scenarios, we defined the northward transport conditions within the region of interest as the latitudinal variability of meteorological and chemical properties experienced by a hypothetical air mass traveling from 40°N to 90°N over a 5-day period. This variability was parameterized into a latitudinal profile, divided into five bins, each representing 10° of northward motion, equivalent s to 1 day of suspension time.

### 2.1.3 Meteorological conditions

Temperature (T) and relative humidity (RH) data were extracted from the ERA5 monthly averaged data on pressure levels between 2010 and 2020 and geographically limited to the region of interest. These global georeferenced reanalysis data are available with a 0.25°x0.25° horizontal resolution and a pressure range of 1000 -1 hPa binned on 37 pressure levels. A detailed description of the ERA5 products is provided by Hersbach et al. (2020). The median of T and RH was calculated for the low-altitude scenarios from the ERA5 pressure levels 1000-950-900-850-800 hPa and for the high-altitude scenarios from the ERA5 pressure levels 600-550-500-450-400 hPa. The mean georeferenced data in the two altitude levels were then organized in 1 longitudinal bin and 5 equally spaced latitudinal bins with a width of 10°N. Each bin represents the median of the 0.25°x0.25° grid in the corresponding altitude and longitude range described before. Daytime and nighttime dramatically change from winter to summer as a function of latitude (Table S1). For the winter ARCTEx simulations (WL and WH), we assumed a decrease of daily light duration from 8 hours on day 1 (40 - 50°N) to 0 hours on day 5 (80-90°N). For the summer ARCTEx simulations (SL and SH), we assumed an increase of daily light duration from 16 hours on day 1 (40 - 50°N) to 24 hours on day 5 (80-90°N). Despite uncertainties, ERA5 temperature data show good agreement with in-situ Arctic observations, with deviations within 6% latitudinally (Pernov et al., 2024) and ~1°C vertically (Graham et al., 2019), supporting its suitability for driving the ARCTEx experimental design. However, the ~40% underestimation of RH suggests limitations in replicating Arctic humidity conditions (Pernov et al., 2024), which may impact the representation of aerosol-phase processes in the chamber.

### 2.1.4 Atmospheric composition

While the source and origin of Arctic BC was often investigated with the variability of sulphate aerosol (Massling et al., 2015), source-partitioning and emission-region of nitrate in is rarely investigated in the Arctic (Moschos et al., 2022). Over the past decades, considering the increase of concentration of nitrate in the last years compared to other inorganic atmospheric species (Zare et al., 2018), nitrogen oxides and nitrate matter might represent, in the future, a not negligible source for BC coatings. Hence, during ARCTEx, nitrogen dioxide ($NO_2$) was chosen as major precursor to form inorganic coating over BC. To investigate the nitrate coating formation, $NO_2$ and BC mixing ratio were extracted from the CAMS (Copernicus Atmosphere Monitoring Service) dataset. These global georeferenced reanalysis data (fourth generation ECMWF global reanalysis; EAC5) are available with a 0.75°x0.75° horizontal resolution and a pressure range of 1000 -1 hPa binned on 25 pressure levels. An overview of the CAMS products is provided by Inness et al. (2019), while details on the aerosol schemes are given in Morcrette et al. (2009) and Bozzo et al. (2017). The representativity of CAMS for Arctic conditions is poorly addressed. However, $NO_2$/BC ratio may be particularly uncertain due to seasonal underestimations of $NO_2$ and an oversimplified treatment of BC's hydrophilic conversion, which may impact transport and removal processes in CAMS scheme (Ryu and Min, 2021; Ding and Liu, 2022). Despite these limitations, CAMS reliably captures large-scale trends and variability, making it a valuable dataset for our experimental design.

In view of the different conditions between the real atmosphere and a simulation chamber, we did not consider the absolute concentration of BC and $NO_2$ but focused on the mass ratio of $NO_2$ over BC ($NO_2$/BC). While $NO_2$ is a direct CAMS product, the ARCTEx BC mass concentration was calculated as the sum of the hydrophilic and hydrophobic components of BC provided by CAMS (Li et al., 2024a). Hence, in our analysis, BC was treated as a single component without distinguishing the hydrophilic and hydrophobic fractions. The $NO_2$/BC mass ratio was calculated on the same pressure levels and region of interest than the ERA5 data described above. The resulting latitude profiles of $NO_2$/BC ratio are listed in Table S1. Although ozone is an essential reactive component in the atmospheric nitrogen chemistry, it was not extracted from CAMS reanalysis data. To simplify the chamber operation, ozone was kept in excess with respect to $NO_2$, without respecting ambient variability. It must be noted that volatile organic compounds which are a by-product of combustion, were simultaneously emitted with BC and injected in the AIDA chamber without active control. As a result, the organic aerosol content in AIDA reflects the specific emissions of the burner and not ambient-like conditions. Therefore, although the experiments primarily targeted the evolution of BC mixing with nitrate coatings, the presence of organic vapors may interact or compete with $NO_2$ during condensation and coating formation, introducing additional complexity to the aging dynamics.

## 2.2 The AIDA aerosol and cloud simulation chamber

The ARCTEx experiments were conducted in July 2022 in the Aerosol Interaction and Dynamics in the Atmosphere chamber (AIDA) at the Karlsruhe Institute of Technology. The chamber was described in detail in previous works (e.g. Möhler et al., 2003b; Saathoff et al., 2003b; Vogel, 2022). Briefly, the AIDA chamber is a cylindrical aluminum vessel with a total volume of 84 $m^3$ equipped with pressure, temperature, humidity, and light control systems. A simplified schematic of the AIDA facility and the instrumental setup is shown in Figure 2. A flow of dry, particle-free synthetic air was continuously injected into the AIDA chamber to compensate for the sampling flow rate and to keep it an overpressure of +1 hPa with respect to ambient. The AIDA chamber is enclosed within a thermal housing, where heat exchangers control the air temperature from 60°C to −90°C. The chamber gas temperature was monitored with 24 fast responding thermocouples orientated in the AIDA chamber volume on vertical and horizontal lines. A mixing fan, positioned 1 m above the bottom of the vessel, ensured that the temperature, gas and aerosol particles were homogeneously mixed within 90 seconds at all times. Water vapor is measured in situ by a tunable diode laser hygrometer (Fahey et al., 2014) and extractively by a dew point mirror hygrometer (MBW373LX, MBW Calibration Ltd.). Based on these measurements, in case of excessive dilution-drying of the chamber, the RH was actively increased by injection of a flow of humidified

synthetic air. AIDA is equipped with a LED light source a to mimic the solar spectrum between 300 nm and 530 nm of wavelength (Vallon et al., 2022). Considering that the emissions of the different LEDs change with temperature, the current of the LED arrays was adjusted at every change of temperature in the chamber to match the spectrum at +20°C. The light source was turned on and off, thereby setting the virtual sunrise, and sunset according to the latitude-dependent irradiation time listed in Table S1. Thanks to its capabilities, the AIDA chamber has been used to investigate a wide range of atmospheric processes including, among the most recent, formation (Gao et al., 2022) and light absorption (Jiang et al., 2024) of secondary organic aerosol, ice nucleating abilities of porous organic aerosol (Wagner et al., 2024), inorganic salts (Bertozzi et al., 2021) and volcanic ashes (Umo et al., 2021), and homogeneous freezing of sulfuric acid (Schneider et al., 2021)

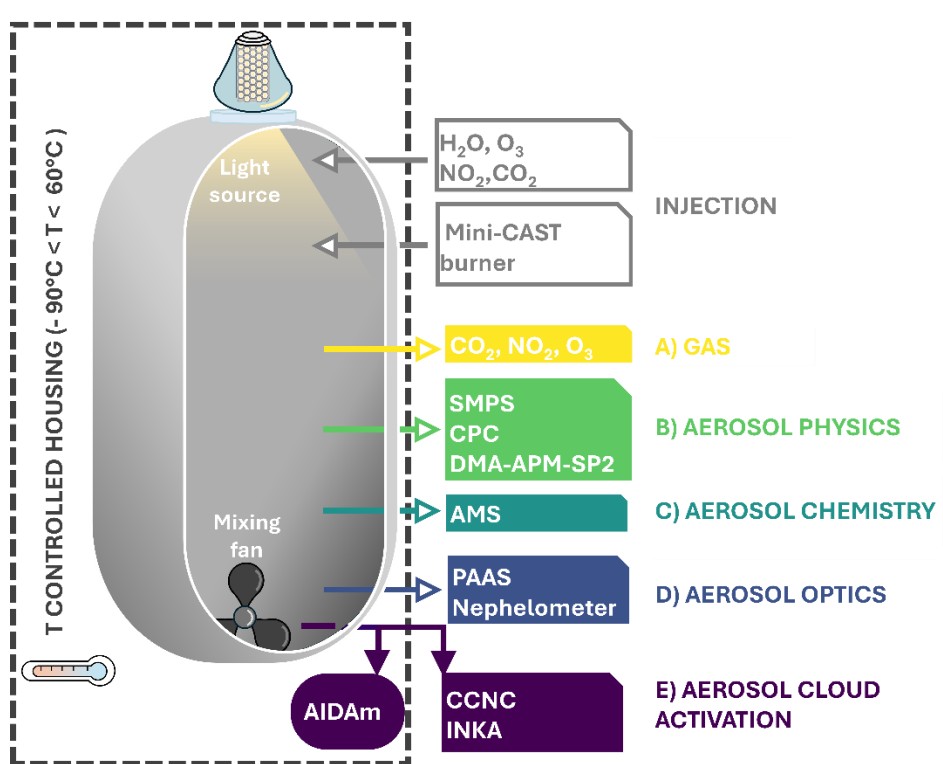

**Figure 2 Schematic representing the AIDA (Aerosol and Dynamics of the Atmosphere) chamber facility. In grey are the injection systems for water vapor ($H_2O$), ozone ($O_3$), nitrogen dioxide ($NO_2$), carbon dioxide ($CO_2$) and soot (mini-CAST burner). In color from top to bottom the measurements packages of : A) gas ($CO_2$, $NO_2$ , $O_3$) ; B) aerosol physics (condensation particle counter - CPC, scanning mobility particles sizer  - SMPS , single particle soot photometer  - SP2 , differential mobility analyzer - aerosol particle mass analyzer - DMA-APM; C) aerosol chemistry (aerosol mass spectrometer - AMS); D) optical aerosol properties ( photoacoustic aerosol absorption photometer - PAAS, nephelometer); E) cloud activation (AIDA-mini - AIDA-m, cloud condensation nuclei counter - CCNC), Ice Nucleation instrument of the Karlsruhe Institute of Technology - INKA).**

## 2.3   Injection and measurement techniques of trace gases

Besides reactive gases, one single injection of carbon dioxide ($CO_2$) was performed at the beginning of each experiment to monitor the dilution caused by the sampled air. While $NO_2$ and $CO_2$ were obtained from gas cylinders (1000 ppm of 99.5 %; Basi Schöberl GmbH), ozone was typically in excess and generated by a silent discharge generator (Semozon 030.2, Sorbios) in pure oxygen (99.9999 %). All gases were injected into AIDA via fluorinated ethylene propylene (FEP) tubing. The duration of the gas injection ranged from few seconds to a minute, depending on the target concentration. The concentrations of ozone, nitrogen dioxide and carbon dioxide were measured with different gas sensors (Figure 2). For

nitrogen dioxide, the cavity phase shift $NO_2$ Analyzer (AS32M Environment S. A.), for ozone the UV photometric ozone monitor (O341M Environment S. A.), and for $CO_2$ the multi-gas infrared analyzer (NGA2000 MLT, Rosemount) were used. All these gas sensors were connected to the chamber through FEP tubes. Sampling from AIDA was performed intermittently using automated valves to reduce the dilution of the AIDA air. More details on gas injection and measurement are given by Saathoff et al. (2009), Gao et al. (2022), and Vallon et al. (2022).

For the benefit of the reader, the look-up table (Table A 1) of all particle properties and their corresponding symbols defined in this, and the following sections is given in the Appendix A.

## 2.4 Injection and measurement techniques of aerosol particles

### 2.4.1 Soot aerosol generation

Soot experiments were already performed in the past at the AIDA chamber using various generation methods to investigate the particles' microphysics (Saathoff et al., 2003b), optical properties (Schnaiter et al., 2003, 2005), CCN activity (Henning et al., 2012), ice nucleation activity (Möhler et al., 2005), and to calibrate-compare measuring techniques (Laborde et al., 2012b). During ARCTEx, soot particles were produced with a mini-CAST burner (series 5200; Jing Ltd Zollikofen, BE, Switzerland) operated with 60 mLPM of propane, 1.55 LPM of oxidation air, 7.5 LPM of $N_2$ and 13 LPM of dilution air. Due to the high internal volume of the AIDA chamber, to reach the target number concentration ($4.5 \times 10^4$ $cm^{-3}$) of soot particles the injection time lasted between 40 and 50 minutes. The functioning principle of the burner and the soot properties are described in detail by Moore et al. (2014). The term 'soot' is commonly used in combustion research to describe carbonaceous particles formed from incomplete combustion (Petzold et al., 2013). To maintain consistency with atmospheric science terminology, we will use 'BC' to describe the evolution of particle properties, except when referencing past studies focused on combustion processes. In addition to BC particles, the mini-CAST burner emits a wide range of volatile and semi-volatile organic compounds (Mamakos et al., 2013; Mason et al., 2020; Daoudi et al., 2023). Although the burner was configured to maximize soot production while minimizing the organic content of the particles (Ess and Vasilatou, 2019), a certain amount of VOCs was inevitably co-emitted and introduced into the AIDA chamber during injection.

### 2.4.2 Concentration and size distribution of total and refractory black carbon aerosol

The total number concentration of aerosol particles having a diameter above 2.5 nm was measured with a condensation particle counter (CPC, model 3776; TSI Inc, St. Paul, MN, USA). Particle size distribution was measured between 13 nm and 750 nm by a scanning mobility particle sizer (SMPS) utilizing a differential mobility analyzer (DMA; TSI Inc, St. Paul, MN, USA) connected to a CPC (3772, TSI Inc, St. Paul, MN, USA). Data were corrected for multiple charge and diffusion with the Aerosol Instrument Manager software.

A single-particle soot photometer (SP2, Droplet Measurement Tehcnologies, Longmont, CO, USA) was used to quantify the mass of refractory black carbon particles (rBC; Petzold et al., 2013). Previous works provide detailed description of the operating principle (e.g., Stephens et al., 2003; Moteki and Kondo, 2010), calibration procedures (e.g., Gysel et al., 2011; Laborde et al., 2012a), and measuring limitations (e.g., Gysel et al., 2012; Zanatta et al., 2021; Schwarz et al., 2022) of the SP2. The incandescent light detector was calibrated with a fullerene soot standard from Alfa Aesar (stock no. 40971, lot no. FS12S011) mass-selected with an aerosol particle mass analyzer (APM; Kanomax Model APM-3600; Ehara et al., 1996). The SP2 provided the number concentration ($N_{rBC}$), the mass concentration ($M_{rBC}$), and the size distribution of rBC particles in the 0.40 – 180 fg mass range, converted to a mass-equivalent diameter ($D_{rBC}$) using a fixed bulk density. Instead of using the canonical density of ambient BC (1800 kg $m^{-3}$; Moteki and Kondo, 2010), we used a bulk density of 1543 kg $m^{-3}$, which is representative of soot produced with a mini-CAST and characterized by a dominant fraction of elemental carbon over the total

carbon (16%; Yon et al., 2015). The rBC mass concentration was corrected for the narrow detection range of the SP2 by calculating the missing mass between 10 and 1000 nm with a lognormal fit of the rBC mass size distribution (Pileci et al., 2021; Zanatta et al., 2018). The missing mass was attributed to the lower detection limit of the SP2 and the relatively small size of the injected particles ( Figure S1 in Section A of the supplementary material). During the first two hours after injection, the undetected mass was estimated to range between 2% and 5%. However, as coagulation-driven growth progressed, the rBC particles fully shifted into the SP2 detection range within approximately six hours, effectively eliminating the initial under-detection issue. The scattering detector was calibrated using monodisperse spherical polystyrene latex (Thermo Fisher Scientific, Waltham, Massachusetts, USA). Due to suboptimal combination of low signal-to-noise ratio of the scattering and position sensitive detectors, and the small optical size of the particles present in the AIDA chamber, it was not possible to quantify the coating thickness with SP2 measurements as proposed by Gao et al. (2007). More details are given in Section 2.4.6.

 and Considering that the concentration measured by the SP2 may be biased by 4% already at an aerosol number concentration of 1000 cm$^{-3}$ (Schwarz et al., 2022), the SP2 was sampling the AIDA air after a dilution system. The dilution was modulated from a factor 100 to a factor 1 depending on the concentration within AIDA to maintain the rBC number concentration measured by the SP2 below 1000 cm$^{-3}$. The dilution factor was quantified as the ratio of the number concentration measured by an undiluted reference CPC and by an auxiliary CPC connected in parallel to the SP2 behind the dilution system. All number and mass concentrations reported hereafter refer to the actual temperature and pressure conditions inside the AIDA chamber (i.e., not normalized to standard conditions) at the time of the measurement.

### 2.4.3   Chemical characterization of non-refractory aerosol particles

A High-Resolution Time-of-Flight AMS (Aerodyne Research Inc., USA) equipped with a PM$_{2.5}$ aerodynamic lens was used to measure the non-refractory PM$_{2.5}$ (NR-PM$_{2.5}$) components including organics, nitrate, sulfate, ammonium, and chloride at a time resolution of 1 minute (DeCarlo et al., 2006; Williams et al., 2013). Chamber air was sampled via a stainless-steel tube with a total flow of 1.1 SLM of which ~84 cm$^3$ min$^{-1}$ were sampled by the AMS. The aerosol particles were then focused into a narrow beam by the PM$_{2.5}$ aerodynamic lens with an effective transmission for particle sizes ranging from ~70 to ~2500 nm (vacuum aerodynamic diameter) and heated by a vaporizer at 600 °C. The resulting vapors are ionized by electron impact (70 eV) and characterized by a time-of-flight mass spectrometer. The AMS ionization efficiency was calibrated by using ~400 nm dried ammonium nitrate aerosol particles. The AMS data was analyzed with the software package SQUIRREL 1.60C and PIKA 1.20C. To account for the effect of particle bouncing loss, chemical-composition-based collection efficiencies (~0.5) were applied to calculate the particle mass concentration (Middlebrook et al., 2012). Elemental analysis of organic aerosol including hydrogen-to-carbon ratio (H:C) and oxygen-to-carbon ratio (O:C) was calculated using the improved ambient method (Canagaratna et al., 2015). In this study, the AMS was used to determine the chemical composition of non-refractory coating material deposited on the BC surface, similar to Cross et al. (2010). Since sulfate, ammonium, and chloride were not introduced into the AIDA chamber directly or indirectly as a byproduct of combustion, their consistently remained below the detection limit. The soot produced by the mini-CAST burner contains minimal organic carbon (Moore et al., 2014), but volatile organic compounds emitted as combustion by-products(Mamakos et al., 2013; Daoudi et al., 2023) may oxidize to form organic coatings (Lim et al., 2019). As a result, organic aerosol was observed during all experiments. In view of these considerations the coating was assumed to consist solely of nitrate and organic components. Additionally, given that changes in particle shape and morphology during the experiments may impact the AMS collection efficiency (Willis et al., 2014), only the mass fractions of nitrate (FM$_{nit}$) and organics (FM$_{org}$) are discussed in this work.

## 2.4.4 Effective density and fractal dimension

The effective density ($\rho_e$) describes the apparent density of aspherical-fractal particles with voids, where the physical diameter does not correspond to the mobility diameter (DeCarlo et al., 2004). The fractal dimension ($d_f$) describes the mass-diameter relationship as a power-law function (Park et al., 2004). During ARCTEx, $\rho_e$ and $d_f$ where quantified by means of DMA-APM system, similar to Park et al. (2004). The particles were first selected with a differential mobility analyzer (DMA; TSI model 3080L; Knutson and Whitby, 1975) according to their mobility diameter ($D_{p\text{-DMA}}$). The particle mass ($m_{p\text{-APM}}$) was calculated as the mean of the normal curve fitted to the mass distribution measured with an Aerosol Particle Mass analyzer (APM; Kanomax Model APM-3600; Ehara et al., 1996) combined with a condensation particle counter (CPC; TSI model 3775). $\rho_e$ was calculated as:

$$\rho_e = \frac{6m_{P-APM}}{\pi D_{P-DMA}^3} \qquad\qquad 1$$

$\rho_e$ measurements were performed before and after the daily change of conditions within AIDA in the mobility diameter range of 100-500 nm with a diameter resolution of 50 nm. Due to varying rates of particle concentration and diameter across experiments, $\rho_e$ calculations were consistently available only in the 150-250 nm mobility diameter range. The uncertainty of $\rho_e$ was quantified as the standard deviation of a normal curve fitted to the mass distribution measured by the APM during all scans performed between 150-250 nm (9 %).

Following Park et al. (2004), the relationship between particle mass and its diameter is described as a power law, where the exponent represents the fractal dimension ($d_f$) and $k_f$ a fit constant.

$$m_{P-APM} = k_f D_{P-DMA}^{df} \qquad\qquad 2$$

The $d_f$ error associated with each scan was calculated as the standard deviation of fit function assuming a 95% confidence interval. Overall, the averaged $d_f$ error was quantified to be 13%, with increasing values at the end of each experiment, when particles number concentration was lower. Considering the various measuring approaches (Cross et al., 2010), the so defined $d_f$, is alternatively called "fractal exponent" (Kim et al., 2009) or "scaling exponent" (Sorensen, 2011) which is analogous although not strictly equivalent to fractal dimension, remaining mostly qualitative (Yon et al., 2015).

## 2.4.5 Mixing state and particle density

The SP2 can be combined with independent aerosol mass quantification techniques to determine aerosol mixing states in both field and laboratory experiments (Sipkens et al., 2021; Naseri et al., 2022). During ARCTEx, the SP2 was used downstream of the DMA-APM system, which was operated in size-mass selection mode. This setup allowed for quantification of the refractory black carbon mass ($m_{rBC\text{-SP2}}$) within the total particle mass selected by the APM ($m_{p\text{-APM}}$). The rBC mass ($m_{rBC\text{-SP2}}$) was calculated by fitting a normal curve to the mass distribution of rBC particles measured by the SP2 after the DMA-APM selection. The mass fraction of rBC ($Fm_{rBC}$) was determined as the ratio of $m_{rBC\text{-SP2}}$ to $m_{p\text{-APM}}$. Using this, the coating mass fraction ($Fm_{coat}$) was calculated as follows:

$$Fm_{coat} = \frac{m_{P-APM} - m_{rBC-SP2}}{m_{P-APM}} \qquad\qquad 3$$

The uncertainty of $Fm_{coat}$ was quantified as the standard deviation of a normal curve fitted to the mass distribution measured by the SP2 during all scans performed between 150-250 nm (15 %).

The average particle density ($\rho_p$) represents the density of the aerosol material. In a double-component system (rBC core and coating), $\rho_p$ was calculated as the mass weighted mean of the density of rBC-material and the coating-material:

$$\rho_P = \rho_{rBC} Fm_{rBC} + \rho_{coat} Fm_{coat} \qquad 4$$

The rBC material density ($\rho_{rBC}$) was assumed to be 1543 kg m$^{-3}$, which is representative of soot produced with a mini-CAST (Yon et al., 2015). Similarly, the contribution of the coating was estimated using a coating material density ($\rho_{coat}$) and its mass fraction. $\rho_{coat}$ was estimated using the AMS-derived relative mass fractions of organic (FM$_{org}$) and nitrate (FM$_{nit}$) material. $\rho_{coat}$ was calculated as function of the relative abundance of organic and nitrate and their corresponding material density as in Equation 5:

$$\rho_{coat} = \rho_{org} FM_{org} + \rho_{nit} FM_{nit} \qquad 5$$

where $\rho_{org}$ represents the organic-material density ($\rho_{org}$ = 1200 kg m$^{-3}$; Lim and Turpin, 2002) and $\rho_{nit}$ represents the nitrate-material density ($\rho_{nit}$ = 1750 kg m$^{-3}$; Pokorná et al., 2022).

We note that this approach assumes a simplified dual-component system with solid and void-free core-shell structure and does not capture the morphology change induced by coating formation. Hence, the calculated $\rho_p$ should be interpreted as an average material density. The ratio $\rho_p/\rho_e$ was used as an indicator of particle compaction. Values near unity suggest a compact particle with internally mixed structure and minimal voids, while lower values indicate a fractal morphology, where internal voids reduce the effective density relative to the material density.

### 2.4.6 Volume equivalent diameter and coating thickness

The volume equivalent diameter ($D_{ve-p}$) is the diameter of a spherical and compact particle with the same volume (DeCarlo et al., 2004), and it is calculated from the particle mass knowing the material density ($\rho_p$; Equation 4) of the total particle as:

$$D_{ve-P} = \sqrt[3]{\frac{6\, m_{P-APM}}{\pi\, \rho_P}} \qquad 6$$

The volume equivalent diameter of the rBC-core was calculated following Equation 7 from the mass of the rBC-core measured with the SP2 ($m_{rBC\text{-}SP2}$) behind the DMA-APM tandem using a fixed material density of 1543 kg m$^{-3}$ (Yon et al., 2015) as:

$$D_{ve-rBC} = \sqrt[3]{\frac{6\, m_{rBC-SP2}}{\pi\, \rho_{rBC}}} \qquad 7$$

For each particle selected by the DMA-APM-SP2 system based on its mobility diameter and mass, it was possible to estimate the volume equivalent coating thickness ($\Delta D_{ve}$; Equation 8). Assuming the particle core is spherical and covered by a spherical, concentric coating layer, $\Delta D_{ve}$ was defined as the half of the difference between the total particle diameter $D_{ve-p}$ and the rBC-core diameter $D_{ve-rBC}$:

$$\Delta D_{ve} = \frac{D_{ve-P} - D_{ve-rBC}}{2} \qquad 8$$

It must be noted that the SP2 is also capable of estimating the coating thickness of rBC-containing particles (Laborde et al., 2012b) by applying the leading edge only fit (LEO-fit) as proposed by Gao et al. (2007). To apply the method with this specific SP2 model, it is essential to obtain signals from the scattering and position-sensitive detectors with a high signal-to-noise ratio for particles having an rBC-

core diameter in the 200-260 nm range (Zanatta et al., 2018). On one side, the number fraction of rBC cores exceeding 200 nm in diameter remained below 1% throughout our experiments, limiting counting statistics. More importantly, even if internal mixing led to increased mobility diameters during the SL and WL experiments, the optical diameters of the rBC-containing particles remained below or close to the detection threshold of the SP2's scattering and position-sensitive detectors. Hence, although the approach proposed here involved the use of several instruments, it represented the only technical solution to determine the coating thickness for small and thinly coated BC-containing particles.

### 2.4.7 Climate-relevant aerosol particle properties

Climate-relevant properties of BC were quantified during ARCTEx with an extended set of instruments (Figure 2). Optical properties such as aerosol absorption and scattering coefficient were quantified at various wavelengths with a photoacoustic aerosol absorption spectrometer (KIT-PAAS; Linke et al., 2016) and a nephelometer (model 3563, TSI Inc, St. Paul, MN, USA; Anderson and Ogren, 1998), respectively. The ability of BC to activate to cloud droplets was quantified with a cloud condensation nuclei counter (CCNC; Droplet Measurement Technologies – DMT, Longmont, CO, USA; Rose et al., 2008). The ice nucleating behavior was quantified at constant temperature and varying supersaturation with the Ice Nucleation Instrument of the Karlsruhe Institute of technology (INKA; Bertozzi et al., 2021) and at the same thermodynamic conditions of AIDA with the AIDA mini (AIDAm; Vogel et al., 2022). While the present work aims to present the ARCTEx project and asses the evolution of fundamental BC properties, the evolution of climate-relevant properties as function of aging will be the topic of study for a companion paper.

### 2.5 Experimental design

In view of the complexity of ARCTEx simulations, we provide a description of the experimental design and schedule. The experiments relied on the assumption that 1 day of experiment corresponded to 10° of northward transport; hence the full transport conditions from 40°N to 90°N were reproduced during 5 days of experiment. The temporal evolution of each experiment was represented with the elapsed time, in hours since the beginning of aging ($t_0$), corresponding to a simulated latitude of 40°N. So that, $t_{24}$ corresponded to 50°N, $t_{48}$ to 60°N, $t_{72}$ to 70°N, $t_{96}$ to 80°N and $t_{120}$ to 90°N. The simulation of northward transport and aging process started at the first virtual sunrise ($t_0$), with the first injection of trace gas and light irradiation. Every 24 hours and 10°N, the chamber conditions such as temperature, humidity, light (irradiation time), and concentration of $NO_2$ and $O_3$ were adjusted to match the transport conditions identified from the reanalysis data. Gas concentrations were continuously monitored using the instruments described in Section 2.3, while aerosol physical and chemical properties were measured as outlined in Section 2.4. The experiment preparation occurred before $t_0$ (negative elapsed time) and included the cleaning and cooling of the chamber ($t_{-24}$) the single injection of BC ($t_{-2}$), its characterization at dark conditions ($t_{-1}$) and the single injection of $CO_2$ ($t_{-1}$). A detailed and idealized schedule for the experimental sequence of operations is given in Table 1.

**Table 1: Schedule for the experimental sequences**

| Elapsed time [h] | Experiment day | Simulated latitude [°N] | Operation |
|---|---|---|---|
| -24 | -1 | - | • Chamber cleaning and conditioning |
| -3 | 1 | - | • Background measurement |
| -2 | 1 | - | • Soot injection |
| -1 | 1 | - | • Dark characterization |
| 0 | 1 | 40-50 | • $O_3$ and $NO_2$ injection<br>• Light on |
| + 22 | 1 | 40-50 | • Temperature adjustment |

| | | | |
|---|---|---|---|
| + 24 | 2 | 50-60 | • $O_3 – NO_2$ injection<br>• Water vapor injection[A]<br>• Light adjustment[B] |
| + 46 | 2 | 50-60 | • Temperature adjustment |
| + 48 | 3 | 70-80 | • $O_3 – NO_2$ injection<br>• Water vapor injection[A]<br>• Light adjustment[B] |
| + 70 | 3 | 70-80 | • Temperature adjustment |
| + 72 | 4 | 80-70 | • $O_3 – NO_2$ injection<br>• Water vapor injection[A]<br>• Light adjustment[B] |
| + 94 | 4 | 80-90 | • Temperature adjustment |
| + 96 | 5 | 80-90 | • $O_3 – NO_2$ injection<br>• Water vapor injection[A]<br>• Light adjustment[B] |
| +110-120 | 5 | 80-90 | • End of experiment |

[A] Since the sampling air was replaced by dry air, the water vapor was replenished every day.
[B] The solar simulator was adjusted to illuminate the chamber with radiation representative for the corresponding latitude.

## 3    Results

This section presents the experimental results in a structured progression. We begin with three preparatory analyses: first, the transport conditions simulated in AIDA (Section 3.1), followed by a characterization of the initial BC particle properties (Section 3.2), and an evaluation of the temporal evolution of particle number and mass concentrations in comparison to real Arctic conditions (Section 3.3). The core of the results focuses on the chemical, morphological, and size evolution of BC during aging (Section 3.4). Finally, we assess the aging timescales of BC under different transport scenarios (Section 3.5).

### 3.1    Northward transport conditions

We start our discussion with the latitudinal profiles of the atmospheric conditions, extracted from ERA-5 and CAMS reanalysis, for the four transport scenarios summer low-altitude (SL), summer high-altitude (SH), winter low-altitude (WL), and winter high-altitude (WH). Figure 3 shows the latitudinal profiles of temperature, relative humidity, and $NO_2$/BC ratio. The temperature decreased with latitude in all experiments, with the strongest gradient at low altitude in summer ($0\ °C < T < 21°C$) and winter ($-21\ °C < T < -9\ °C$). A weaker gradient but marked low temperature was observed at high altitude in summer ($-41\ °C < T < -35\ °C$) and winter ($-58\ °C < T < -52\ °C$). Relative humidity increased with latitude up to 80% in the low altitude scenarios, while more stable and dryer conditions were observed at high altitude ($RH < 65\ \%$). These temperature and humidity profiles are consistent with general Arctic conditions, which typically feature stronger latitudinal cooling and more humid conditions near the surface compared to higher altitudes  (Przybylak, 2016). The longer atmospheric lifetime of $NO_2$ during winter (Levy II et al., 1999) may be responsible for the higher $NO_2$/BC ratio in winter than in summer, in both low and high altitude scenarios. Although elevated $NO_2$ relative to BC concentration is a prerequisite for nitrate coating formation, the production pathways in the Arctic are strongly influenced by extreme environmental factors such as low temperatures (Alexander et al.; 2020) and limited sunlight (Schaap et al., 2004), which affect both the efficiency and timing of nitrate formation. As concluded in the recent AMAP (2021) report, the limited horizontal and vertical coverage of $NO_2$ measurements does not allow for further comparison with ambient data.

To verify the representativity of the AIDA simulations, we compared the reanalysis "setpoints" with the daily mean values observed in the chamber (Figure 3). For all the considered scenarios, we were able to reproduce the latitudinal profile of temperature with a relative precision of 1%, without introducing any

bias. The difference between the simulated and measured RH showed higher variability than temperature.
This difference was not systematic and varied from a maximum overestimation of + 27% (day 1 of WH) to – 20% (day 1 of SL). Despite it, we were able to reproduce in the AIDA chamber the latitudinal RH gradient extracted form ERA-5. The mismatch in RH may have affected gas-phase and aqueous-phase chemistry processes influencing the formation of coating-precursors and coating material. As shown by the high values of the standard deviation of the $NO_2$/BC ratio measured in the AIDA chamber, the control
of $NO_2$ concentration proved to be the most complicated. This technical difficulty led to non-systematic bias during the various experiments. As an example, we introduced contrasting bias in the low altitude scenarios. While the high $NO_2$/BC in SL might promote a higher degree of internal mixing in AIDA compared to CAMS, the depleted level of NO2 might have hindered coating formation and internal mixing in WL. In summary, atmospheric conditions extracted from ERA-5 and CAMS revealed complex
and heterogeneous transport conditions, which were well reproduced, day-by-day, in the AIDA chamber. Nonetheless, discrepancies between reanalysis and measured $NO_2$ levels might accelerate or hinder coating formation in a non-systematic way across the experiments. The mean of these atmospheric conditions extracted from the reanalysis and AIDA dataset are listed in Section B of the supplementary material (Table S1, Table S2).

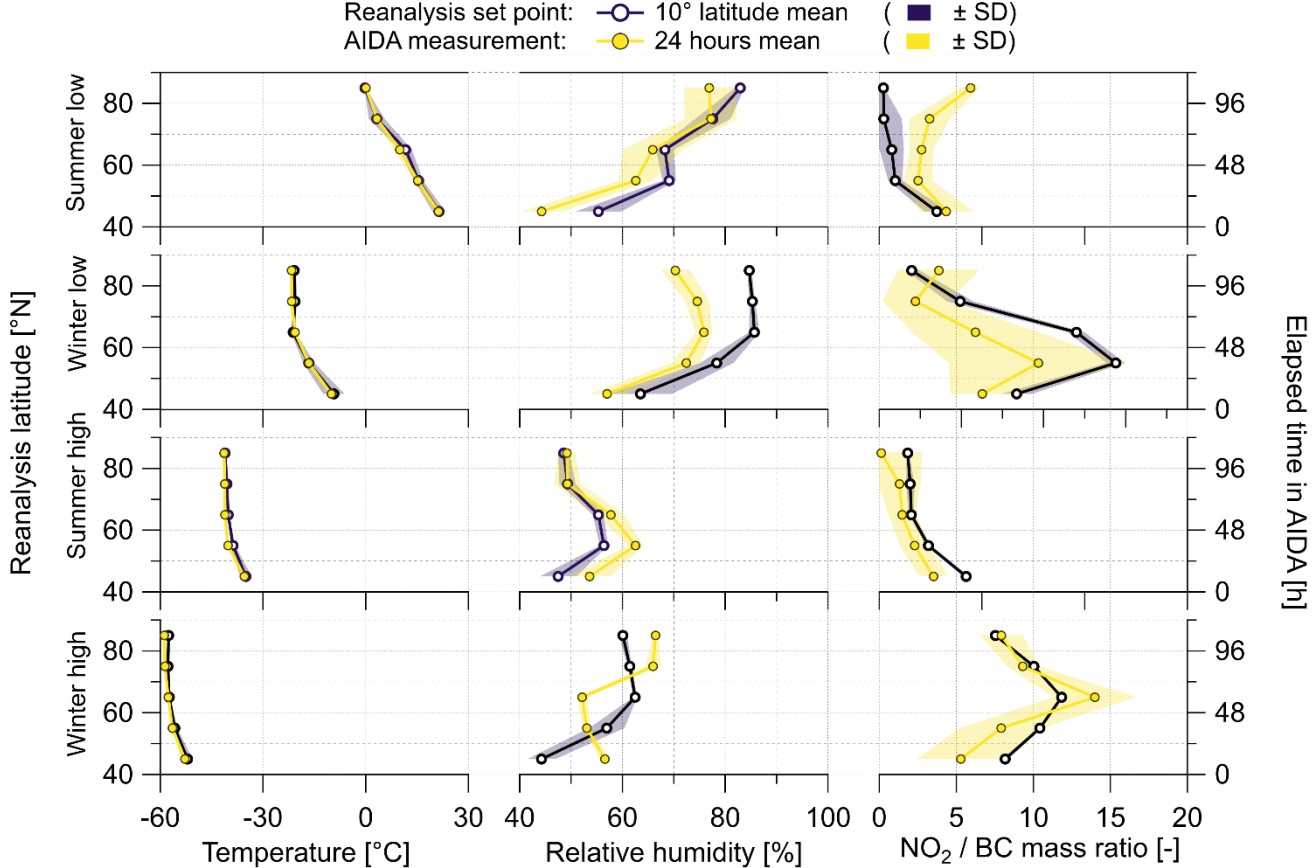

**Figure 3 Left axis: Latitudinal profiles of temperature and relative humidity extracted from ERA-5 and of $NO_2$/BC mass ratio extracted from CAMS in the region of interest (40-90°N and 60-140°E). Mean and standard deviation (SD) calculated for equidistant latitude bands 10° wide. Right axis: temporal variability of temperature, relative humidity and $NO_2$/BC ratio measured in the AIDA**
**chamber. Mean and standard deviation (SD) calculated over 24 hours.**

## 3.2 Characterization of fresh Mini-CAST soot

The physical characterization of fresh soot produced with the mini-CAST was performed before $t_0$ (Table
1) and is fully described in Section C of the supplementary material. Key findings are summarized as follows. The SMPS-measured geometric mean diameter ranged from 80 nm (SL) to 104 nm (SH) (Figure

S 2a), consistent with prior studies using diffusion flame burners with similar oxidative airflows (Ess et al., 2021; Maricq, 2014; Moore et al., 2014; Rissler et al., 2013). Figure S 2b shows that the diameter-dependent decrease in effective density is comparable to precedent studies with diffusion-flame soot (Cross et al., 2010; Rissler et al., 2013; Ess et al., 2021). The low fractal dimensions ($d_f$ = 2.01-2.21) of the generated particles (Figure S 2c) matched those reported previously for diffusion-flame soot (Ess et al., 2021; Rissler et al., 2013; Heuser et al., 2024), diesel soot (Olfert et al., 2007) and premix-flame soot (Cross et al., 2010). The rBC mass fraction ($F_{rBC}$) varied between 95% and 75% in agreement with elemental to total carbon ratios from premixed (Cross et al., 2010) and diffusion (Schnaiter et al., 2006; Ess et al., 2021; Heuser et al., 2024) burners. Note that rBC and elemental carbon should be compared cautiously (Pileci et al., 2021). The soot produced during ARCTEx featured complex geometry and a dominant refractory mass fraction. While our soot properties fell within the range reported in previous studies, the unique experimental design and setup of ARCTEx may account for the differences observed with earlier experiments.

## 3.3 Evolution of particle number and mass concentrations

In this section, we briefly discuss the variability of number and mass concentrations measured with the SP2. A target concentration of 4.5 x $10^4$ cm$^{-3}$ at t$_{-1}$ was chosen to ensure the suspension of at least a few hundred particles per cm$^3$ after 120 hours. This accounted for a dilution factor of 25-30% per day, which was driven by the sampling flow of the gas and aerosol measuring instruments. Overall, N$_{rBC}$ remained above 300 cm$^{-3}$ until 110-115 hours after t$_0$, enough to perform a full characterization of its chemical and physical properties. In terms of mass concentration, the rBC levels decreased during all scenarios from a maximum of 15 µg m$^{-3}$ at t$_0$ to a minimum of 0.5 µg m$^{-3}$ at t$_{115}$ (Figure 4). Although the concentrations in AIDA were adjusted to ensure the extended duration of the experiments and to approximate near-real conditions (based on the NO$_2$/rBC mass ratio), it is insightful to compare these values with previous rBC mass concentrations measured by SP2 in various environments. Due to the utilization of similar rBC concentrations in low- and high-altitude scenarios for experimental reasons, it is evident that the ARCTEx concentrations do not accurately reflect the natural decreasing trend of rBC with altitude as observed across Europe (McMeeking et al., 2010; Dahlkötter et al., 2014; Zanatta et al., 2020) and in the high and low Arctic (Schulz et al., 2019; Juрányi et al., 2023; Zanatta et al., 2023). The M$_{rBC}$ observed during the first 24 hours (40-50°N) was representative of polluted Asian megacities (~20 µg m$^{-3}$; Li et al., 2023; Yu et al., 2020). M$_{rBC}$ on the second day (50-60°N) reflected polluted events over continental Europe (~ 4 µg m$^{-3}$; Laborde et al., 2013; Yuan et al., 2020). Although decreasing due to particle dilution, M$_{rBC}$ remained above typical Arctic background values of northern Finland (0.12 µg m$^{-3}$; Raatikainen et al., 2015) and European Arctic haze (0.04 µg m$^{-3}$; Zanatta et al., 2018).

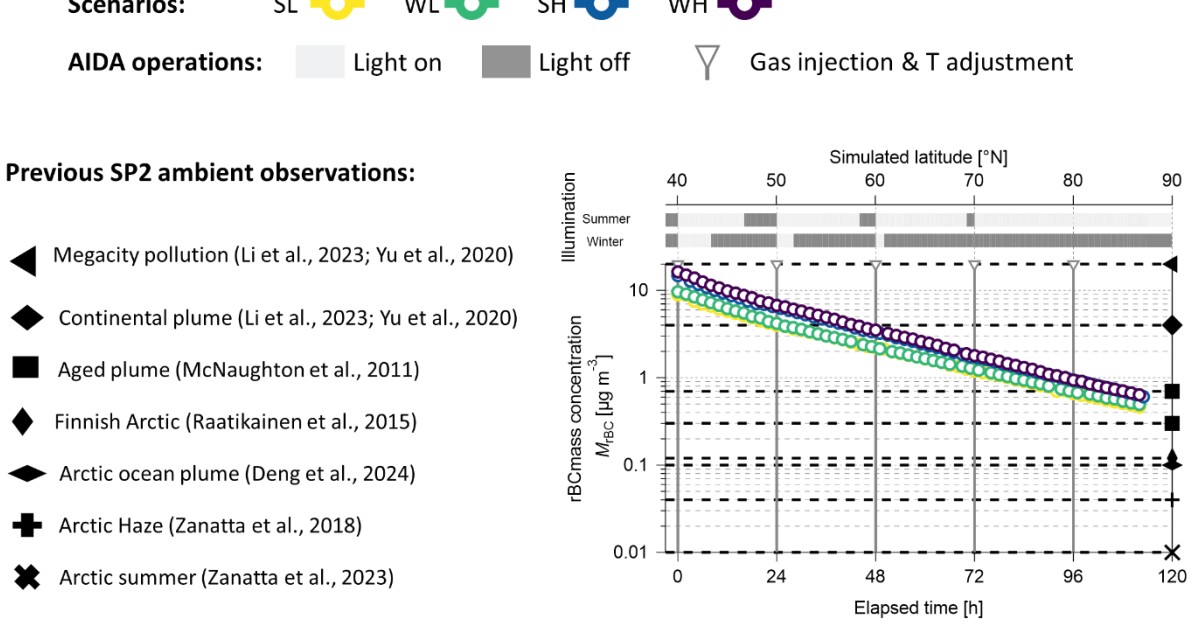

**Figure 4 Temporal and latitudinal evolution of rBC mass concentration measured with the single particle soot photometer SP2 between 0.40–180 fg. Concentrations normalized to the gas temperature and pressure of the AIDA chamber at the time of measurement. ARCTEx scenarios: summer low-altitude (SL), winter low-altitude (WL), summer high-altitude (SH), winter high-altitude (WH). Concentration compared with previous ambient M*rBC* observations.**

### 3.4 Evolution of BC morphology and mixing state during aging

In this section we describe the temporal evolution of morphology and mixing composition of particles in the 150-250 nm diameter range as described in Section 2.4.4 and Section 2.4.5. These properties include: the particle effective density ($\rho_e$; Figure 5a), the particle fractal dimension ($d_f$; Figure 5b), the mass fraction of coating material ($Fm_{coat}$; Figure 5c), the total particle density ($\rho_p$; Figure 5d), the effective to particle density ratio ($\rho_e / \rho_p$ ; Figure 5e) and the volume equivalent coating thickness ($\Delta D_{ve}$; Figure 5f). The mass fraction of nitrate ($FM_{nit}$) and organic ($FM_{org}$) is shown in Figure 6. Excluding $FM_{nit}$ and $FM_{org}$, the other properties described in this section were calculated from the DMA-APM and DMA-APM-SP2 scans. These scans were performed before and after the virtual midnight, resulting in a low temporal resolution.

### 3.4.1 Organic driven aging of BC in summer low-altitude transport

During the SL experiment, $\rho_e$, $d_f$ and $Fm_{coat}$ increased significantly from $t_0$ (40°N) to $t_{96}$ (80°N), indicating a profound change in the morphology and mixing of BC (Figure 5a-b-c). Although $\rho_e$ increased by a factor 2 and $Fm_{coat}$ reached 50% already in the first 24 hours (40-50°N), clear indications for particle compaction were only observed after $t_{48}$, when $d_f$ increased to values between 2.5 and 2.8 (60-70°N). At this stage, $\rho_e$ varied within 614-982 kg m$^{-3}$ and $Fm_{coat}$ within 56-63%. Organics dominated the overall composition of coatings with $FM_{org}$ constantly above 90% (Figure 6a). Considering that $O_3$ decreased at a faster rate compared to the other scenarios, the long irradiation time might promote the photolysis of ozone to form OH radicals. In turn, OH may oxidize volatile organic carbon, emitted by the burner (Mamakos et al., 2013; Daoudi et al., 2023), to secondary organic material (Lim et al., 2019). This reaction pathway efficiently competed with the oxidation of $NO_2$ to nitric acid and its condensation to nitrate coating during day time (Alexander et al., 2009). The low $NO_2$/BC ratio and prolonged irradiation time prevented the typical nitrate night-formation pathways via $NO_3$ reaction with volatile organic carbon (Ng et al., 2017) or with $NO_2$ to form $N_2O_5$ followed by hydrolysis on the BC surface (Chang et al., 2011; Alexander et al., 2020).

Given that organic material has a lower density than nitrate and BC, the nearly constant $FM_{org}$ combined with an increase in $Fm_{coat}$ resulted in a steady decrease in particle density, contrasting with the increase observed in the other experiments (Figure 5d). Nonetheless, $\rho_e$ (1350 kg m$^{-3}$) approached $\rho_p$ (1299 kg m$^{-3}$) at the end of the experiment at $t_{96}$ (Figure 5e). Since the effective density and material density are equivalent for spherical particles, a 4% difference suggests that BC particle in SL attained sphericity. It must be noted that uncertainties in the AMS measurements (Liu et al., 2007) or assumptions regarding the highly variable organic density (Kostenidou et al., 2007) might influence this result. In view of this and considering the coating composition and the overarching assumption of sphericity, the volume equivalent coating thickness (Figure 5f) monotonically increased from ~10 nm at $t_{24}$ (40-50°N) to 41 nm at $t_{96}$ (80°N).

### 3.4.2 Nitrate driven aging of BC in winter low-altitude transport

During the WL experiment, $\rho_e$ rapidly increased from ~200 kg m$^{-3}$ to approximately 1500 kg m$^{-3}$ within 48 hours of aging (60°N), after which it remained constant until $t_{96}$ (80°N), indicating the formation of spherical particles (Figure 5a). This compaction towards sphericity was confirmed by $d_f$ values approaching 3 after 48 hours, although some unphysical values above 3 were observed due to limited DMA-APM scans (Figure 5b). These values might reflect the ± 13% error associated with the fitting method used to derive $d_f$ (Section 2.4.4). The increase in $Fm_{coat}$ from 23% to 87% within the first 48 hours, peaking at 97% by $t_{96}$, suggests that winter low-altitude conditions can lead to fully encapsulated

BC particles within two days, or before reaching 60°N (Figure 5c). In contrast to the SL scenario, the higher $NO_2$/BC ratio favored nitrate formation over organic coatings, with a rapid increase in the nitrate mass fraction ($FM_{NO3}$) during irradiation periods (Figure 6b). This increase suggests that nitrate formed via OH radical oxidation to nitric acid, which then condensed onto the BC, with limited competition from secondary organic aerosol formation as in the case of SL. The complete darkness during the last three experimental days likely promoted nighttime nitrate formation while preventing photolysis of gas precursors ($NO_3$; Dorn et al., 2013) and nitrate (Reed et al., 2017; Ye et al., 2016). As a result, $FM_{NO3}$ increased sharply in the first 7 hours to 60%, reaching 88% after 48 hours (60°N) and 93% after 114 hours (~90°N). Opposite to summer, the increasing dominance of nitrate led to an increase in particle density to ~1700 kg m$^{-3}$ (Figure 5d), agreeing within 10% with the final effective density (Figure 5e). This supports the hypothesis of BC becoming fully encapsulated by a coating already after 48 hours of aging. Assuming a concentric core-shell structure, the coating thickness was initially thin (<10 nm) in the first 24 hours (50°N) but increased to 45 nm at $t_{48}$ (60°N), and ultimately reached the highest value of all ARCTEx scenarios with 54 nm at $t_{96}$ (80°N; Figure 5f).

### 3.4.3 Slower aging during high altitude transport

Despite the different chemistry and timescale, both SL and WL scenarios led to significant BC aging, producing fully encapsulated, spherical particles. In contrast, particle morphology and mixing showed a weaker evolution during high altitude transport in both summer and winter. The effective aerosol particle density $\rho_e$ increased by a factor of 2.24 during SH and by a factor of 1.25 during WH from $t_0$ to $t_{96}$, a substantially smaller change compared to the low altitude experiments ( Figure 5a). Although some compaction was observed, especially in SH, $d_f$ and $Fm_{coat}$ never exceeded 2.4 and 35%, respectively, in both scenarios (Figure 5b and Figure 5c). This indicates a very small degree of aging with limited impacts on the morphology of the BC particles at very low temperatures from mid latitudes to the Arctic. Under these conditions of persistent asphericity and negligible mixing, the coating chemistry and thickness quantification are highly uncertain. However, nitrate dominated the composition of coating (Figure 6c and Figure 6d), $\rho_e/\rho_p$ remained well below unity (Figure 5e) and $\Delta D_{ve}$ remained below 10 nm until $t_{96}$ or 80°N (Figure 5f). Hence, high altitude transport may slow the aging process of BC, resulting in completely different BC properties in the Arctic region as function of altitude. This aging reduction, was likely linked to low temperatures, which may slightly reduce the yield of ozone photolysis (Matsumi et al., 2002), shorten the lifetime of $HNO_3$ (Dulitz et al., 2018), and minimize the oxidation of volatile organic compounds to secondary organic aerosol coatings (Saathoff et al., 2009; Tillmann et al., 2009). This hypothesis is reinforced by the near-constant depletion rates of $NO_2$ and $O_3$, irrespective of light conditions, suggesting limited photochemical and nocturnal reactivity at temperatures below -30 °C. Although the formation of secondary organo-nitrate aerosol is found to be temperature dependent (Gao et al., 2022), due to the lack of comprehensive gas-phase (nitric acid, OH radical, volatile organic compounds) and particle-phase (absolute quantification of nitrate, organic and organo-nitrate compounds) chemical speciation, we cannot quantify the coating yields or attribute them to specific reaction pathway.

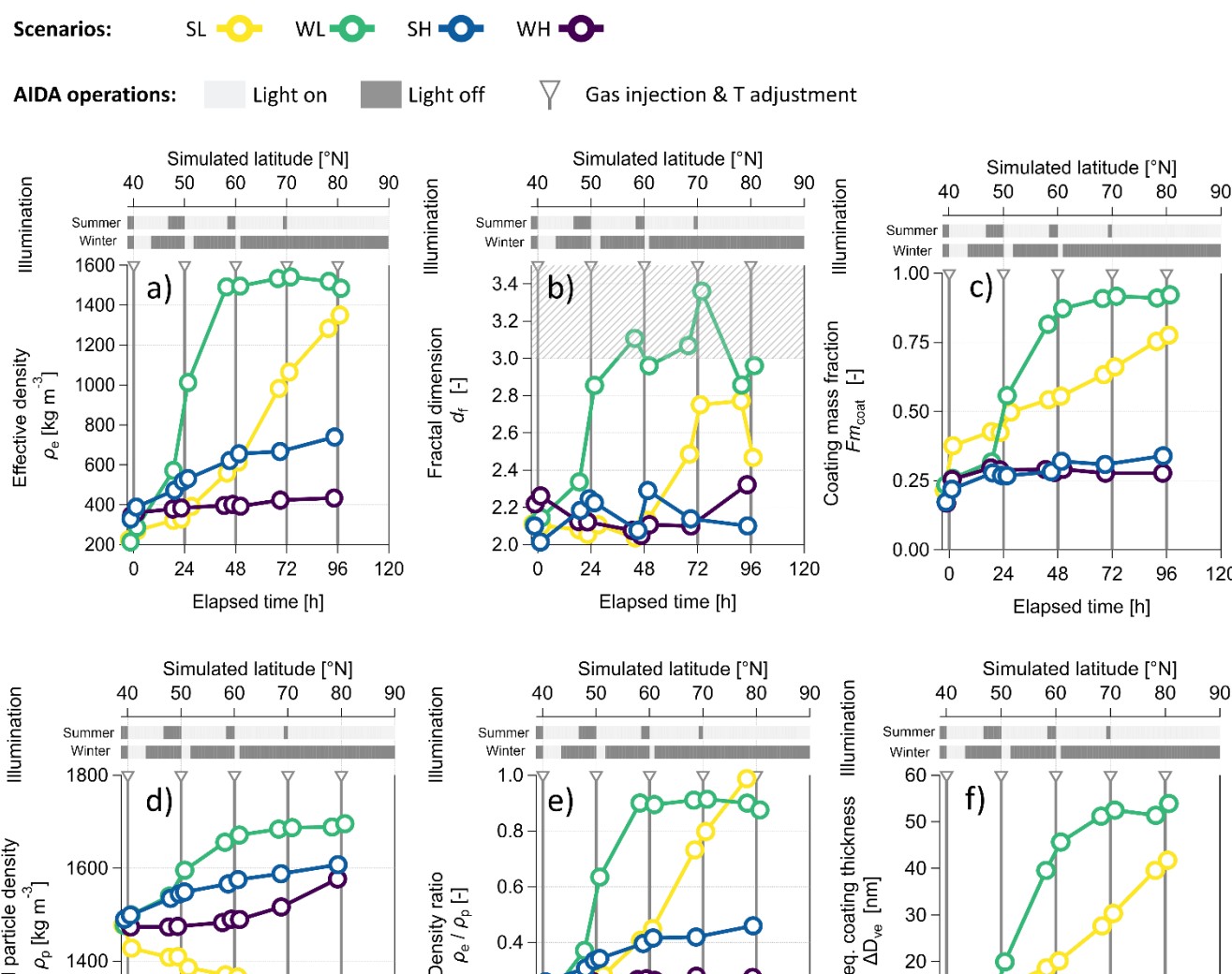

**Figure 5 Temporal and latitudinal evolution of rBC physical properties. a) Particle effective density, $\rho_e$; b) Particle fractal dimension, $d_f$; c) Coating mass fraction, $Fm_{coat}$; d) total particle density, $\rho_p$; e) Effective to particle density ratio, $\rho_e/\rho_p$ ; f) volume equivalent coating thickness, $\Delta D_{ve}$. All properties were calculated for particles with a mobility diameter between 150-250 nm. ARCTEx scenarios: summer low-altitude (SL), winter low-altitude (WL), summer high-altitude (SH), winter high -altitude (WH).**

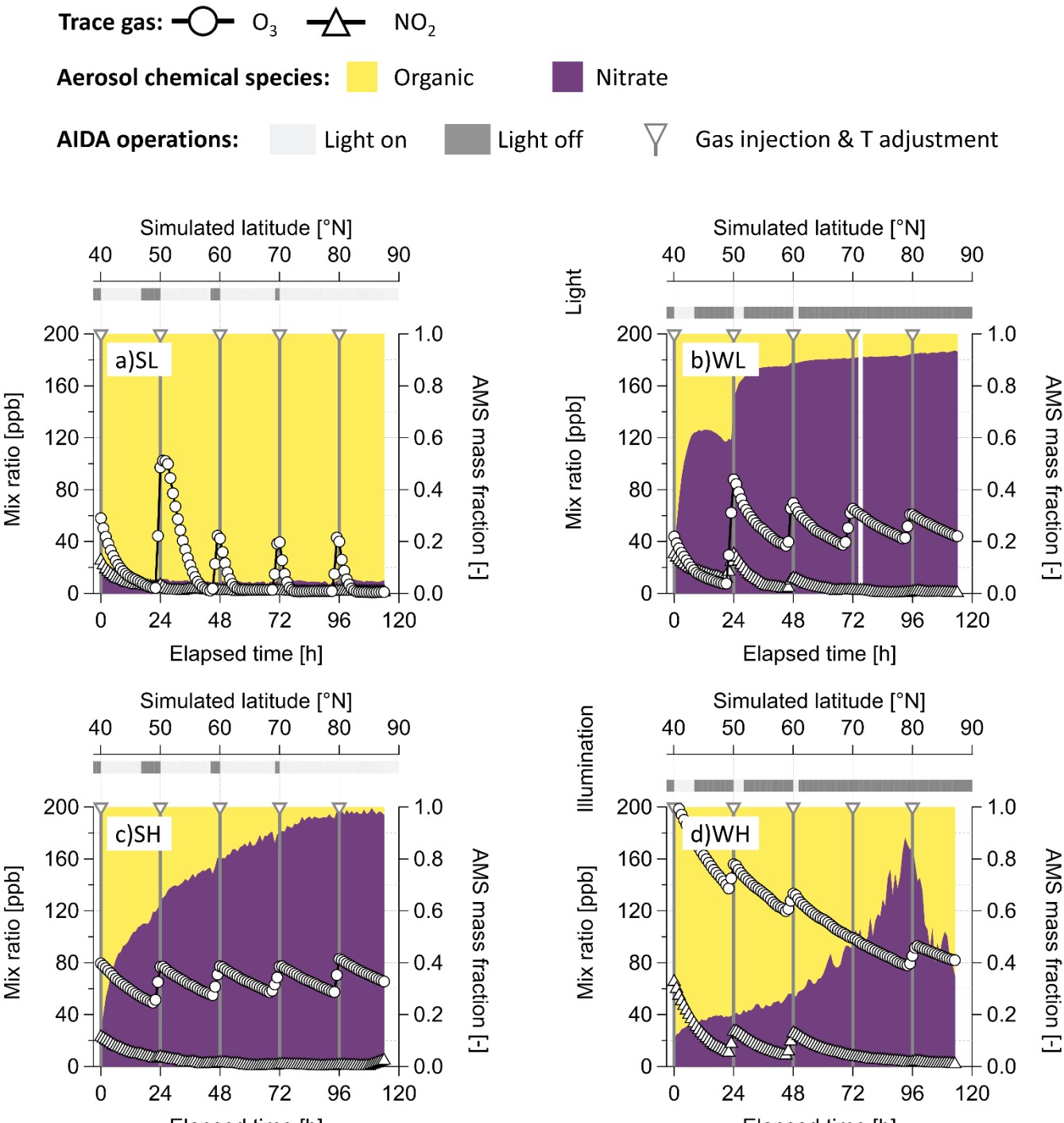

**Figure 6 Temporal and latitudinal evolution of the chemical composition of rBC coatings (organic and nitrate) and volume mixing ratio of ozone ($O_3$ in circles) and nitrogen dioxide ($NO_2$ in triangles) during the ARCTEx scenarios: a) summer low-altitude, SL; b) winter low-altitude ,WL; c) summer high-altitude, SH; d) winter high-altitude, WH.**

### 3.4.4 Mass closure of total aerosol

The rBC mass concentration measured by the SP2 was converted to total aerosol mass concentration ($M_{p-SP2}$) using the particle-by-particle coating mass fraction. Similarly, the number size distribution from the SMPS was converted into a mass size distribution by considering the size-dependent effective density to calculate the corresponding mass concentration ($M_{p-SMPS}$). A detailed description of the methodology is available in Section D of the supplementary material. By comparing these two variables, we aim to assess the accuracy of the $Fm_{coat}$ and $\rho_e$ measurements.

A strong correlation between the two mass concentrations was observed across all scenarios (Figure S 3), with the lowest value in WL ($R^2 = 0.84$) and the highest in SL ($R^2 = 0.99$). However, significant variability was observed across scenarios: the $M_{p-SMPS}$ / $M_{p-SP2}$ ratio ranged from a maximum of 1.6 (WH) to a

minimum of 0.89 (WL), with an ARCTEx average of 1.14 (Figure 7a). This average indicates an overall overestimation of $M_{p\text{-SMPS}}$ (or $M_{p\text{-SP2}}$ underestimation), ranging from 0% in WL to 25% in WH. The $M_{p\text{-SMPS}}$ / $M_{p\text{-SP2}}$ ratio was compared with effective density (Figure 7b), fractal dimension (Figure 7c), and coating mass fraction (Figure 7d), but no clear correlations were found. Notably, the absence of coating in WH ($\rho_e < 500$ kg m$^{-3}$; $d_f < 2.4$; $Fm_{coat} < 0.3$) was linked to the highest discrepancies between the two methods. In contrast, little variability was seen in WL when particles were dense, spherical, and thickly coated ($\rho_e > 1000$ kg m$^{-3}$; $d_f > 2.8$; $Fm_{coat} > 0.7$).

Considering the instrumental uncertainties of the SMPS and the SP2, as well as the quantification uncertainties of $m_{p\text{-APM}}$ ($\pm9\%$) and $m_{rBC\text{-SP2}}$ ($\pm15\%$), the two mass concentrations align well within the instrumental uncertainties, confirming the reliability of the results presented in this work.

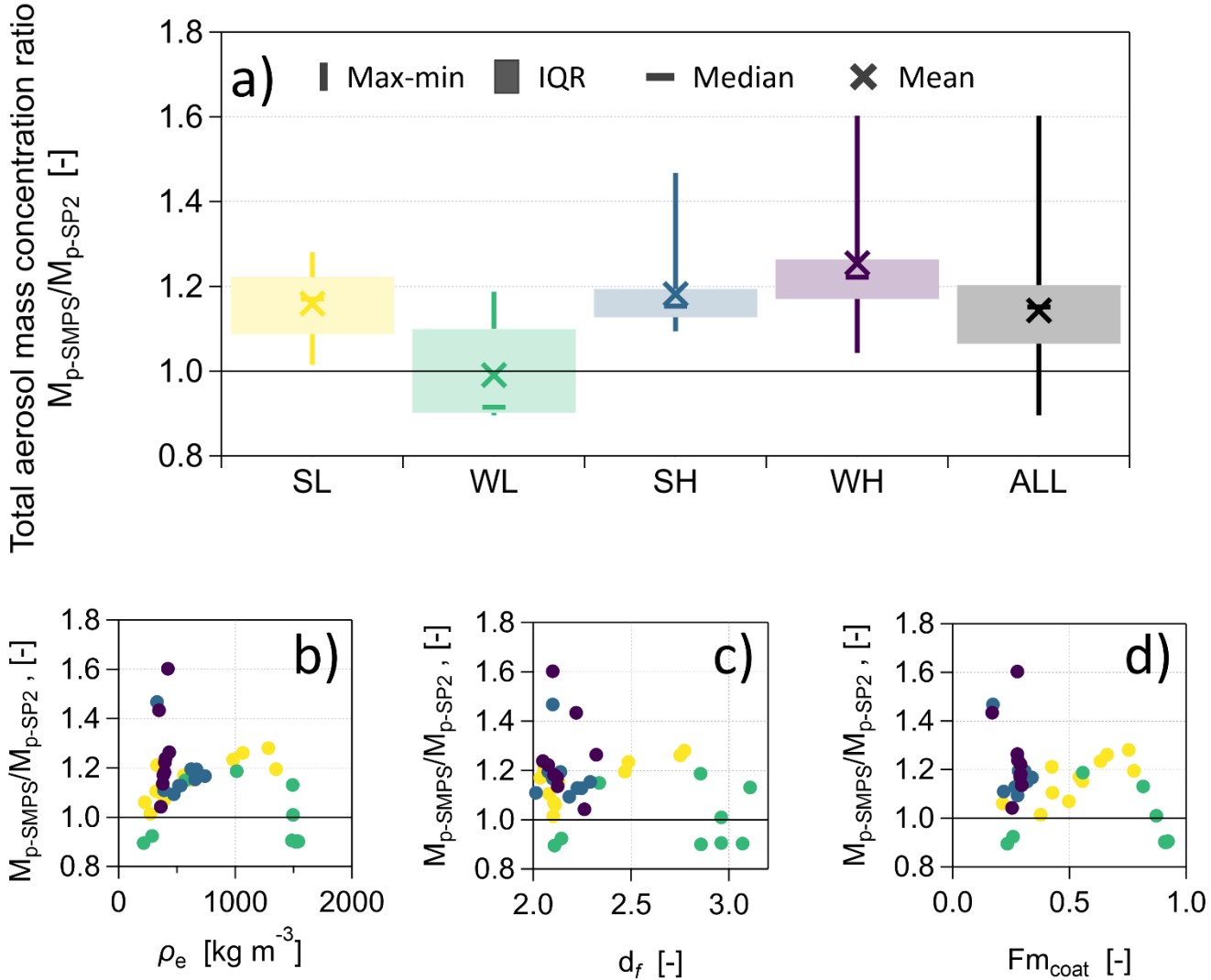

Figure 7 Comparison between the total aerosol mass concentration derived from the SMPS ($M_{p\text{-SMPS}}$) and SP2 ($M_{p\text{-SP2}}$). a) Statistic of the $M_{p\text{-SMPS}}$ / $M_{p\text{-SP2}}$ ratio for each scenario. Correlation of $M_{p\text{-SMPS}}$ / $M_{p\text{-SP2}}$ ratio with: b) effective density, $\rho_e$; c) fractal dimension, $d_f$; d) coating mass fraction, $Fm_{coat}$. All properties calculated for particles with a mobility diameter between 150-250 nm.

### 3.4.5 Aging impact on particle diameter

During all experiments, particle diameter evolved with time following different timescales (Figure S4). The measurement of rBC core diameter (SP2) and total particle diameter (SMPS) where used to assess intra-coagulation and coating formation, respectively. While here we provide a short summary, all details are given in Section E of the supplementary material. Excluding WL, the rBC core diameter increased

exponentially (a decreasing growth rate at decreasing concentration), leading to a net diameter growth between 65% and 85% by the experiment's end (90°N). In WL, diameter growth shifted from exponential to linear at the coating onset. The concurrent increase in fractal dimension may have reduced active surface area, limiting collisions and slowing coagulation. Despite prior studies (e.g. Schnaiter et al., 2003; Maricq, 2007), coagulation during the transition from external to internal mixing remains poorly characterized. While coagulation increases diameter while maintaining external mixing (Naumann, 2003), coating deposition first compacts and reduces the size of BC (Schnaiter et al., 2003; Bambha et al., 2013; Yuan et al., 2020), then increases diameter with coating thickening (Li et al., 2017). These processes were clearly visible in the low-altitude scenarios. The gradual formation of thin coating was associated with a gradual collapse of BC 's ramified structure during SL and allowed to observe a slow but steady reduction of mobility diameter. The rapid attainment of sphericity in WL, did not allowed to observe compaction but rather the quick formation of thick coatings already every coating stage. A similar step-wise growth was observed during medium-duration aging experiments in AIDA (Schnaiter et al., 2005).

## 3.5 Varying aging timescales as function of transport pathway

Generally, the aging timescale defines the suspension time required for a BC particle to transition from a hydrophobic to a hydrophilic state. Global models parametrize this conversion in several ways. The most simplified approach in bulk aerosol models is to consider a fixed ageing timescale (Koch et al., 2009). Other aging schemes, used in modal aerosol modules, include a coating thickness threshold made up of a variable number of mono-layers (Liu et al., 2016). Considering the large discrepancies between observations and simulations, more detailed treatment of size, morphology and mixing is implemented in the modules of global models (e.g. (Matsui, 2016; Chen et al., 2024; Jin et al., 2025). In this section, we aim to determine the aging time and latitude scales of BC as a function of its degree of internal mixing, rather than hygroscopicity, quantified as the ratio of coating material to the rBC core in mass ($Rm_{coat}$). Sedlacek et al. (2022) showed that the increase in $Rm_{coat}$ is proportional to the age of pollution plumes, following a first-order growth model reaching a maximum of 20 within 1 day of suspension time. We first address the evolution of $Rm_{coat}$ during the aging scenarios (Figure 8a). While the highest $Rm_{coat}$ values were recorded during the WL ($Rm_{coat}$ = 12.3 at $t_{96}$) and SL ($Rm_{coat}$ = 3.70 at $t_{96}$), the high-altitude scenarios demonstrated slower aging, with maximum $Rm_{coat}$ values remaining below unity. According to Sedlacek et al. (2022), the ARCTEx $Rm_{coat}$ values would correspond to real plume age shorter than a day. The slower mixing, can be attributed to the absence of coagulation aging processes within AIDA compared to ambient conditions, where such processes promote quicker aging and thicker coatings (Matsui et al., 2013). To provide an aging time scale we analyzed the relationship between $Rm_{coat}$ and the fractal dimension ($d_f$; Figure 8b) as well as the effective to particle density ratio ($\rho_e/\rho_p$; Figure 8c). Both properties are indicative of the sphericity or compaction of aerosol particles and were found to increase exponentially with $Rm_{coat}$, reaching an asymptotic maximum. These results reinforce the significant role of coating formation in influencing the sphericity of BC particles (Leung et al., 2017; Yuan et al., 2020). We aimed to identify a critical $Rm_{coat}$ at which fractal BC restructures into more compact shape. According to previous works that studied the variability of aerosol shape with fractal dimension (Olfert et al., 2007; Wang et al., 2017; Leskinen et al., 2023) and particle density (Rissler et al., 2014), we extrapolated $Rm_{coat}$ corresponding to a $d_f$ of 2.8 and to a $\rho_e/\rho_p$ of 0.9 based on the exponential equations shown in Figure 8. The so-derived critical $Rm_{coat}$ of 3.24 and 3.64 marked the transition point at which fractal BC aggregates approached a spherical shape ($d_f$ = 2.8) and their observed material density ($\rho_e/\rho_p$ = 0.9), respectively.

Next, we derived the aging timescale necessary to reach the critical coating mass for each scenario. We applied a Hill equation (Weiss, 1997; Goutelle et al., 2008) to model the non-linear relationship between $Rm_{coat}$ and BC age, forcing the upper (maximum $Rm_{coat}$ observed during WL) and lower (initial $Rm_{coat}$ from each scenario) boundaries along with a 95% confidence interval (Figure 8a, dashed lines). Compared to a more generic sigmoid equation, the Hill function allowed capturing the sudden increase of $Rm_{coat,}$ in the early phase of the aging. From this fitted function, we derived the aging timescale corresponding to the critical $Rm_{coat}$ range (3.24–3.64, black crosses in Figure 8). The critical $Rm_{coat}$ was reached within the

experimental time for both low-altitude scenarios, with the aging times quantified as 39-40 hours for WL
and 92-98 hours for SL, corresponding to latitudes of 53-57°N and 78-81°N, respectively. On the contrary, the coating formation rate was slower in the high-altitude experiments, with aging time-scales exceeding the experiment duration, extrapolated to be 60-70 days for SH and over 600 days for WH. It is important to note that these aging timescales are subject to considerable uncertainty due to the fitting curve, as indicated by the wide confidence bands in Figure 8a. Our experimental results reinforce the
recent findings of Fierce et al. (2025), who highlighted the inadequacy of fixed ageing timescales in models. Their work confirms that ageing rates are regionally and seasonally dependent, as observed in the ARCTEx scenarios, significantly affecting simulated BC concentrations, particularly in the Arctic. Moreover, the altitudinal and seasonal ageing patterns shown in Figure 8 reflect ambient variability and lead to heterogeneous impacts on the hygroscopic and optical properties of BC (Jin et al., 2025)


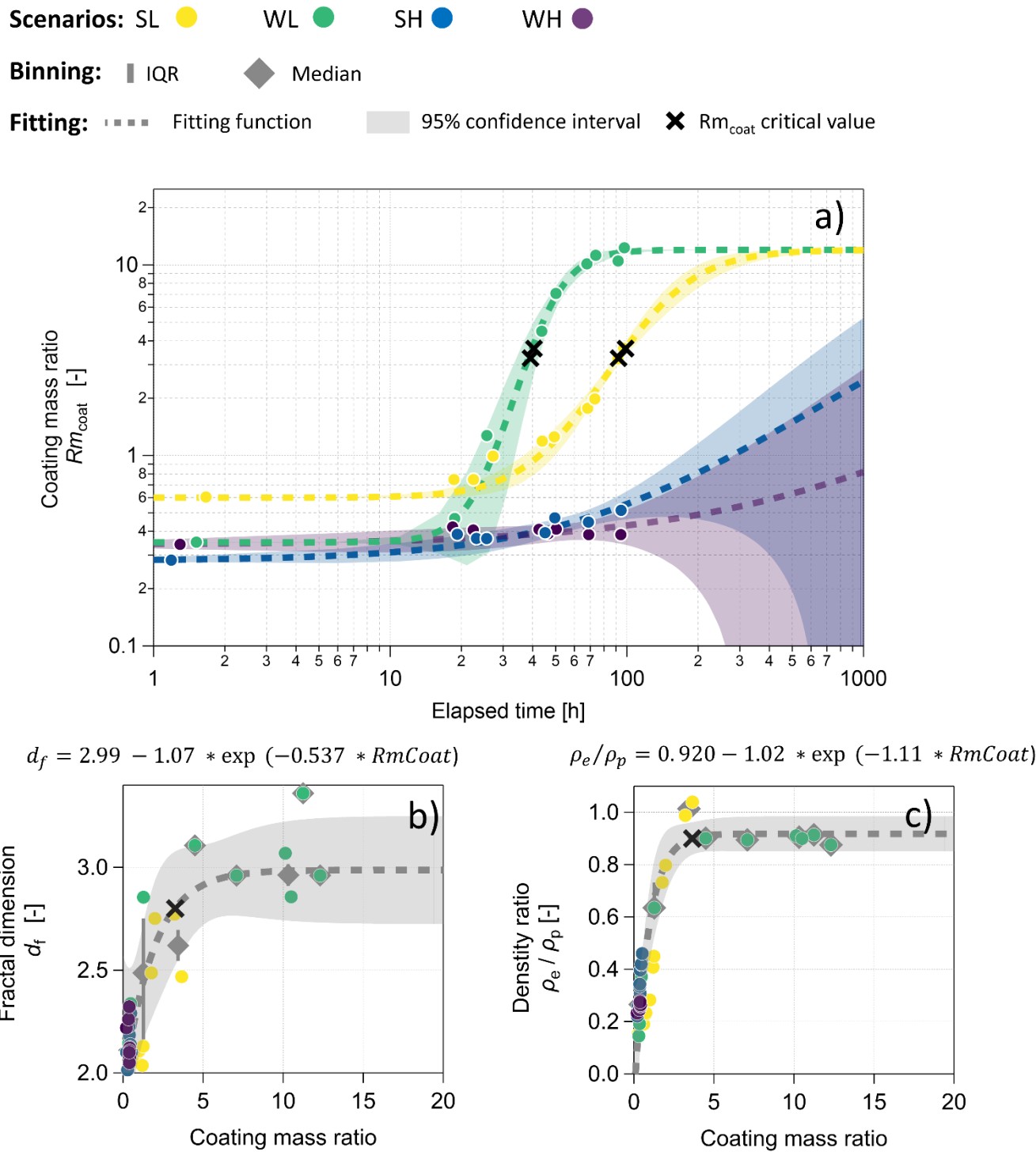

$$d_f = 2.99 - 1.07 * \exp\left(-0.537 * RmCoat\right)$$

$$\rho_e/\rho_p = 0.920 - 1.02 * \exp\left(-1.11 * RmCoat\right)$$


**Figure 8 Comparison between the coating mass-ratio (Rm$_{coat}$) with: a) experiment elapsed time, b) fractal dimension, $d_f$; c) effective to particle density ratio, $\rho_e/\rho_p$. All properties calculated for particles with a mobility diameter between 150-250 nm. ARCTEx scenarios: summer low-altitude (SL), winter low-altitude (WL), summer high-altitude (SH), winter high -altitude (WH). See text for**
**details regarding the fitting functions.**

## 4    Conclusions

The ARCTEx campaign aimed to quantify the aging timescale of BC particles during Arctic transport by simulating quasi-real conditions derived from reanalysis data in the AIDA chamber. The chamber successfully reproduced the meteorological and chemical conditions of four distinct transport scenarios,
with a focus on nitrate and organic, aging and sustained aging experiments for up to 115 hours.

The fundamental physical properties of BC showed distinct temporal evolution depending on the simulated environmental conditions. Temperature proved to be a critical factor controlling the evolution of BC properties until the end of the transport (~110 – 120 hours, 90°N). High-altitude experiments conducted at low temperatures (-59 °C < T < -35 °C) exhibited minimal changes in mixing state and

morphology, characterized by a final coating mass fraction of 25 – 30%, coating thickness of 5–10 nm, fractal dimension of 2.0–2.4, and effective density of 400–800 kg m$^{-3}$. In contrast, low-altitude experiments conducted at higher temperatures (-22 °C < T < 21 °C) resulted in significant internal mixing and compaction. These scenarios yielded a final coating mass fraction of 78–97%, coating thickness of 40–60 nm, fractal dimension of 2.8–3.0, and effective density of 1300–1500 kg m$^{-3}$. Notably, summer

conditions with positive temperatures, extended irradiation, and limited nitrogen oxide availability favored the formation of organic coatings compared to winter, where nitrate coatings were dominant.

This study successfully quantified the time required to transform fresh, externally mixed, fractal BC particles into aged, internally mixed, and compact particles across the four scenarios. The fastest aging was observed in winter at low altitudes, where particles became fully coated within 39–40 hours after

emission, corresponding to latitudes of 53–57°N. In summer at low altitudes, the aging timescale was slightly longer, ranging from 92–98 hours and corresponding to latitudes of 78–81°N. In contrast, high-altitude transport scenarios were characterized by aging timescales exceeding the experiment duration (120 hours) and reaching the northernmost latitude (90°N), independently from the season.

For the first time, reanalysis data were used to drive extended aging experiments in a simulation chamber

replicating Arctic transport, enabling the evaluation of aging effects on BC's fundamental properties. Hence, our work indicates that the aging timescale and impacts on fundamental BC properties vary dramatically as a function of altitude and season of transport. These experimentally derived timescales and transformation pathways provide crucial input for constraining BC aging parameterizations in models, helping to bridge the gap between laboratory-scale studies and real-world Arctic conditions. In

forthcoming work, we will discuss the implications of this temporal evolution on climate-relevant properties such as light absorption and activation as cloud droplets and ice crystals. Together, these studies aim to support the modelling community in improving the representation of BC aging processes in transport simulations, ultimately enhancing the accuracy of climate projections.

# Appendix A

**Table A 1 List of abbreviations, described property, unit, type (M = measured; C = calculated merging different measurements; A = assumed), and measuring instrument or source.**

| Symbol | Property | Unit | Type | Instrument /source |
|---|---|---|---|---|
| $d_f$ | Particle fractal exponent | - | C | DMA-APM |
| $D_{p\text{-}DMA}$ | Particle mobility diameter | nm | M | SMPS |
| $D_{Np\text{-}GM}$ | Geometric mean of the number size distribution of total particles | nm | M | SMPS |
| $D_{NrBC\text{-}GM}$ | Geometric mean of the number size distribution of rBC | nm | M | SP2 |
| $D_{ve\text{-}p}$ | Particle volume equivalent diameter | nm | C | DMA-APM-SP2 / AMS |
| $D_{ve\text{-}rBC}$ | rBC particle volume equivalent diameter | nm | M | SP2 |
| $Fm_{coat}$ | Coating mass fraction | - | C | DMA-APM-SP2 |
| $Fm_{rBC}$ | rBC mass fraction | - | C | DMA-APM-SP2 |
| $FM_{nit}$ | Nitrate mass concentration fraction | - | M | AMS |
| $FM_{org}$ | Organic mass concentration fraction | - | M | AMS |
| $GF_{Dp}$ | Growth factor of $D_{Np\text{-}GM}$ | - | C | SMPS |
| $GF_{DrBC}$ | Growth factor of $D_{NrBC\text{-}GM}$ | - | C | SP2 |
| $GR_{Dp}$ | Growth rate of $D_{Np\text{-}GM}$ | - | C | SMPS |
| $GR_{DrBC}$ | Growth rate of $D_{NrBC\text{-}GM}$ | - | C | SP2 |
| $k_f$ | Fractal constant | - | C | DMA-APM |
| $M_{rBC}$ | rBC particles mass concentration | ng m$^{-3}$ | M | SP2 |
| $M_{p\text{-}SMPS}$ | Mass concentration of total particles derived from SMPS | | C | SMPS |
| $M_{p\text{-}SP2}$ | Mass concentration of total particles derived from SP2 | | C | SP2 |
| $m_{p\text{-}APM}$ | Single particle mass | fg | M | APM |
| $N_{rBC}$ | Number concentration of rBC particles | cm$^{-3}$ | M | SP2 |
| $\Delta D_{ve}$ | Volume equivalent coating thickness | nm | C | DMA-APM-SP2 / AMS |
| $\rho_{BC}$ | BC core material density | kg m$^{-3}$ | A | Yon et al., 2015 |
| $\rho_e$ | Effective density | kg m$^{-3}$ | C | DMA-APM |
| $\rho_{coat}$ | Coating material density | kg m$^{-3}$ | C | AMS |
| $\rho_p$ | Particle density | kg m$^{-3}$ | C | DMA-APM-SP2 / AMS |
| $\rho_{nit}$ | Nitrate material density | kg m$^{-3}$ | A | Pokorná et al., 2022 |
| $\rho_{org}$ | Organic material density | kg m$^{-3}$ | A | Lim and Turpin, 2002 |
| $Rm_{coat}$ | Ratio of coating over rBC core mass | | C | DMA-APM-SP2 |


# Author contributions

MZ conceived the project, conducted the microphysics measurements, and wrote the manuscript. OM, HS, RW, FV and AH contributed to the design of the experiments. PB, YG, NU, FV performed the ice nucleating particles measurements. CL and MS performed the optical measurements. YH, FJ, YL performed the aerosol chemistry measurements. PG and PL provided the single particle soot photometer. All authors contributed equally to the writing of the manuscript.

# Financial support

The ARCTEx project is funded by the Deutsche Forschungsgemeinschaft (DFG, German Research Foundation, grant no. 457895178 (Accessed on 24-07-2025). MZ was partially supported by ITINERIS project (IR0000032), the Italian Integrated Environmental Research 760 Infrastructures System (D.D. n. 130/2022 - CUP B53C22002150006) Funded by EU - Next Generation EU PNRR- Mission 4 "Education and Research". Open Access funding enabled and organized by Project DEAL.

# Data availability

The reanalysis data used to characterize the meteorological transport conditions in this study are publicly available through the Copernicus Climate Data Store: ERA5 monthly averaged data on pressure levels from 1940 to present; DOI: 10.24381/cds.6860a573 (Accessed on 24-07-2025).  The reanalysis data used to characterize the chemical transport conditions in this study are publicly available through the Atmosphere Data Store: CAMS global reanalysis (EAC4) monthly averaged fields; DOI: 10.24381/fd75fff2 (Accessed on 24-07-2025). All data acquired during the ARCTEx experiments in the AIDA chamber, are publicly available on the KITopen RADAR repository, DOI: 10.35097/7gh3j1jzabrzkcx9  (Accessed on 24-07-2025).

# Competing interests

The authors declare no conflict of interest.

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
