# Peer review of "AIDA Arctic transport experiment (part 1): simulation of northward transport and aging effect on fundamental black carbon properties"

_Aerosol Research, 2025_

## Author Response (AR1)

**ar-2025-12: "AIDA Arctic transport experiment (part 1): simulation of northward transport and aging effect on fundamental black carbon properties"**

**Answers of the authors to Reviewer#1**

While the reviewer's comments are given in **black bold**, our answers are given below in blue letters.
Additionally, we added the changes made in the revised manuscript in *blue italic letters.*

**The work "AIDA Arctic transport experiment (part 1): simulation of northward transport and aging effect on fundamental black carbon properties " is well presented. The temporal evolution of BC and the influence of aging processes on the morphological transformations of BC throughout their atmospheric transport to the Arctic regions are systematically investigated through controlled experimental simulations employing realistic atmospheric parameters in the AIDA simulation chamber. The results demonstrate that aging processes coupled with Arctic-bound atmospheric transport induce transformative modifications to the physicochemical properties of BC and the methodology presented demonstrates the pressing need to bridge the gap between laboratory-based measurements and real-world scenarios. The compositional variability of organic and nitrate coating on BC particles during transport under summer and winter conditions is presented for the first time.**

We would like to thank the referees for their detailed and constructive comments, which helped us to improve our manuscript. Here we provide some major considerations. Several reviewers noted an insufficient discussion on the ambient representativity of transport conditions, soot generation and coating species. This has now been addressed in more detail in both the Methods and Results sections, representing the major modification to the text. Reviewers also identified inconsistencies in the use of acronyms and abbreviations. These have been reviewed and corrected throughout the manuscript text and figures. The reviewer's specific comments are addressed as it follows.

**The discrepancy between AIDA measurements and ERA-5 data (Fig. 3) should be discussed. Could it be due to differences in methodologies used or varying in ambient conditions.**

Since this point was raised by multiple reviewers we addressed it in detail.

a. Besides representing a major air transport pattern, the region of interest was chosen also because it shows a limited longitudinal variability in surface temperature, proving relatively stable atmospheric conditions. This was made clear at the end of Section 2.1.1 "Region of interest":
   *... "Although other efficient transport patterns exist, this specific continental area is characterized by a reduced temperature variability (Przybylak, 2016), ensuring more homogeneous atmospheric conditions compared to the Atlantic and Pacific transport pathways."...*

b. To provide a better context to the reanalysis results, a short sub chapter was added in Section 3.1 "Northward transport conditions:

   *... "These temperature and humidity profiles are consistent with general Arctic conditions, which typically feature stronger latitudinal cooling and more humid conditions near the surface compared to higher altitudes (Przybylak, 2016). The longer atmospheric lifetime of $NO_2$ during winter (Levy II et al., 1999) may be responsible for the higher $NO_2$/BC ratio in winter than in summer, in both low and high altitude scenarios. Although elevated $NO_2$ relative to BC concentration is a prerequisite for nitrate coating formation, the production pathways in the Arctic are strongly influenced by extreme environmental factors such as low*

*temperatures (Alexander et al.; 2020) and limited sunlight (Schaap et al., 2004), which affect both the efficiency and timing of nitrate formation. As concluded in the recent AMAP (2021) report, the limited horizontal and vertical coverage of NO₂ measurements does not allow for further comparison with ambient data." ...*

c. Comparability of simulated and measured conditions. Considering that the estimated and observed temperatures were in close agreement, we focused on relative humidity and NO2/BC ratio. The mismatch in RH may have affected gas-phase and aqueous-phase chemistry processes controlling the formation of coating-precursors and coating material. The most affected scenario would be WL, when RH values were underestimated by 10-15% during the entire duration of the experiment. The effect of NO2/BC mismatch would be, however, dominant for the coating formation potential. In this regard, we might have introduced contrasting bias in the low altitude scenarios. We have added a short paragraph at the end Section 3.1 "Northward transport conditions" to explicitly discuss the implications of these discrepancies. The modified text in section 3.1 now reads:

*... "The mismatch in RH may have affected gas-phase and aqueous-phase chemistry processes influencing the formation of coating precursors and coating material. As shown by the high values of the standard deviation of the NO₂/BC ratio measured in the AIDA chamber, the control of NO₂ concentration proved to be the most complicated. This technical difficulty led to non-systematic bias during the various experiments. As an example, we introduced contrasting bias in the low altitude scenarios. While the high NO₂/BC in SL might promote a higher degree of internal mixing in AIDA compared to CAMS, the depleted level of NO₂ might have hindered coating formation and internal mixing in WL. In summary, atmospheric conditions extracted from ERA-5 and CAMS revealed complex and heterogeneous transport conditions, which were well reproduced, day-by-day, in the AIDA chamber. Nonetheless, discrepancies between reanalysis and measured NO₂ levels might accelerate or hinder coating formation in a non-systematic way across the experiments."* ...

d. Figure 3 was modified to accommodate the comments done by different referees. The modifications include: i) Addition of shading to represent the standard deviation of both reanalysis and measurements; ii) different colouring to improve readability; iii) adjustment of axis labels and legend.

[Figure]

*Figure 3 Left axis: Latitudinal profiles of temperature and relative humidity extracted from ERA-5 and of NO₂/BC mass ratio extracted from CAMS in the region of interest (40-90°N and 60-140°E). Mean and*

*standard deviation (SD) calculated for equidistant latitude bands 10° wide. Right axis: temporal variability of temperature, relative humidity and $NO_2$/BC ratio measured in the AIDA chamber. Mean and standard deviation (SD) calculated over 24 hours.*

**Technical comments**

**In the abstract norward should be replaced with northward**

The mistake was corrected.

**replace twentyfour with 24 in line 200**

The issue was addressed.

**rewrite the sentence after Eq. 7 line 375**

The Sentence was indeed awkward. It was modified to improve readability as:

*… "For each particle selected by the DMA-APM-SP2 system based on its mobility diameter and mass, it was possible to estimate the volume equivalent coating thickness ($\Delta D_{ve}$; Equation 7). Assuming that the particle core is spherical and surrounded by a concentric, spherical coating layer, $\Delta D_{ve}$ was defined as the half of the difference between the total particle diameter $D_{ve\text{-}P}$ and the rBC-core diameter $D_{ve\text{-}rBC}$:" …*

**References**

Alexander, B., Sherwen, T., Holmes, C. D., Fisher, J. A., Chen, Q., Evans, M. J., and Kasibhatla, P.: Global inorganic nitrate production mechanisms: comparison of a global model with nitrate isotope observations, Atmospheric Chemistry and Physics, 20, 3859–3877, https://doi.org/10.5194/acp-20-3859-2020, 2020.

AMAP: AMAP Assessment 2021: Impacts of Short-lived Climate Forcers on Arctic Climate, Air Quality, and Human Health, 2021.

Levy II, H., Moxim, W. J., Klonecki, A. A., and Kasibhatla, P. S.: Simulated tropospheric NO : Its evaluation, global distribution and individual source contributions, Journal of Geophysical Research: Atmospheres, 104, 26279–26306, https://doi.org/10.1029/1999JD900442, 1999.

Przybylak, R.: The Climate of the Arctic, Springer International Publishing, Cham, https://doi.org/10.1007/978-3-319-21696-6, 2016.

Schaap, M., Van Loon, M., Ten Brink, H. M., Dentener, F. J., and Builtjes, P. J. H.: Secondary inorganic aerosol simulations for Europe with special attention to nitrate, Atmos. Chem. Phys., 4, 857–874, https://doi.org/10.5194/acp-4-857-2004, 2004.

**ar-2025-12: "AIDA Arctic transport experiment (part 1): simulation of northward transport and aging effect on fundamental black carbon properties"**

**Answers of the authors to Reviewer#2**

While the reviewer's comments are given in **black bold**, our answers are given below in blue letters. Additionally, we added the changes made in the revised manuscript in *blue italic letters*.

**The manuscript "AIDA Arctic transport experiment (part 1): simulation of northward transport and aging effect on fundamental black carbon properties" is a nicely written report on the ARTEX experiment. The authors used the AIDA simulation chamber to mimic the atmospheric processes that age black carbon particles during transport from the mid-latitudes to the Arctic region. In Part 1, they report changes in the physical properties of the aerosol (e.g. diameter, fractal dimension, coating) and relate them to atmospheric conditions. The manuscript is clear and well written, but some sections need more explanation. These parts are summarised below, with some other technical suggestions.**

We would like to thank the referees for their detailed and constructive comments, which helped us to improve our manuscript. Here we provide some major considerations. Several reviewers noted an insufficient discussion on the ambient representativity of transport conditions, soot generation and coating species. This has now been addressed in more detail in both the Methods and Results sections, representing the major modification to the text. Reviewers also identified inconsistencies in the use of acronyms and abbreviations. These have been reviewed and corrected throughout the manuscript text and figures. The reviewer's specific comments are addressed as it follows.

**44: "primarily" >> BC is exclusively emitted from combustion sources.**

In this case, we intended the BC to be a primary aerosol, we agree the sentence was misleading. Now the text reads:

... *"Black carbon (BC) is a primary carbonaceous aerosol emitted by combustion processes"* ...

**109: "Siberian open fires" >> In fact, many of the BC particles transported to the Arctic come from forest fires. On the other hand, burning biomass releases not only BC but also huge amounts of organic aerosol (OA), which is mostly mixed internally with BC. The OA emissions from the burner used in the experiment are negligible compared to the combustion of biomass. OA ageing and secondary OA contribute to changes in the physical and optical properties of the aerosol mixture. I miss the inclusion of organic components in the experiment. You should mention the importance and role of OA and explain why you focused mainly on nitrate-based coatings.**

Siberian biomass burning is an important source of black carbon and organic aerosol during fire season. However, our experiment did not mean to focus on forest fires emissions, but rather on the interaction of elemental carbon rich soot with nitrate species. Nitrate coating has been largely ignored in the past years, and gained more and more importance in view of the increase of nitrate aerosol in the Arctic, which is in contrast to decreasing sulphate and black carbon concentrations (AMAP, 2021). The lack of knowledge on BC-nitrate interaction, was the main reason to design ageing experiments focussing on nitrogen-based species. In this context, ARCTEx did not aim to reproduce biomass burning events. In fact, the choice of our region of interest (Eurasia), besides being an efficient transport corridor, was justified by relatively homogeneous meteorological conditions compared to

other Arctic transport patterns over the Atlantic and Pacific. This homogeneity highly simplified the operation of the AIDA-chamber and increased the representativity of ambient conditions. We do realize, however, that our choice was not properly justified in the main text:

- The text in Section 2.1.1 was modified to better describe the importance of homogeneous meteorological conditions within the region of interest:

  *... "This sector is associated with intense export of anthropogenic (Backman et al., 2021) and natural (McCarty et al., 2021) black carbon emissions. Although other efficient transport patterns exist, the continental Eurasian sector is characterized by a reduced temperature variability (Przybylak, 2016), ensuring more homogeneous atmospheric conditions compared to the Atlantic and Pacific transport pathways. This uniformity is critical for the design of chamber experiments, as it minimizes external variability and facilitates the representation of the ambient conditions, thereby reducing the operational complexity of the AIDA-chamber." ...*

- The text in section 1 "Introduction" was modified to better explain the relevance of nitrate aerosol in the Arctic region:

  *... "BC variability in the Arctic was often associated with co-emitted sulfate aerosol from anthropogenic sources (e.g. Massling et al., 2015) and organic aerosol from biomass burning events (e.g. Moschos et al., 2022), while its correlation with nitrate was mostly ignored (AMAP, 2021). Similarly, chamber studies focused on the evolution of BC properties as function of internal mixing with sulfate (e.g. Möhler et al., 2005; Khalizov et al., 2009; Henning et al., 2012) and organics (e.g. Lefevre, 2019; Wittbom et al., 2014). As a consequence, the impact of BC-nitrate internal mixing on fundamental and climate relevant properties remained poorly assessed (Yuan et al., 2020). Internal mixing of BC with nitrate species becomes particularly important in the Arctic region, where nitrate aerosol concentration has been increasing since the '80s despite an overall reduction of nitrogen oxides emissions (AMAP, 2021). The same report underlined how few studies had focused on nitrate aerosol in the Arctic, introducing a knowledge gap on the sources of its precursors, its formation mechanisms and its interaction with other atmospheric species such as BC." ...*

**111: The NO2/BC ratio was fixed according to the CAMS data set. However, the OA/BC ratio was specific to the burner. This means that your experiment represents evolution of the inorganic coating only. This is not a problem, but you should state this fact explicitly. Otherwise, based on the introduction, the reader is expecting a comprehensive simulation of the Arctic transport of BC particles emitted by wildfires.**

- To specify that we did not have control on the OA/BC ratio with AIDA, we introduced a dedicated subparagraph at the end of Section "2.1.4 Atmospheric composition" which reads:

  *... "It must be noted that volatile organic compounds, which are a by-product of combustion, were simultaneously emitted with BC and injected in the AIDA chamber without active control. As a result, the organic aerosol content in AIDA reflects the specific emissions of the burner and not ambient-like conditions. Therefore, although the experiments primarily targeted the evolution of BC mixing with nitrate coatings, the presence of organic vapours may interact or compete with NO₂ during condensation and coating formation, introducing additional complexity to the ageing dynamics." ...*

- More details on the production of organic compounds by the mini-CAST burner were added at the end of Section 2.41 "Soot aerosol generation":

  *... "In addition to BC particles, the mini-CAST burner emits a wide range of volatile and semi-volatile organic compounds (Mamakos et al., 2013; Mason et al., 2020; Daoudi et al., 2023). Although the burner was configured to maximize soot production while minimizing the organic content of the particles (Ess and Vasilatou, 2019), a certain amount of VOCs was inevitably co-emitted and introduced into the AIDA chamber during injection." ...*

**183-184: In the results you did not separate BC into hydrophilic and hydrophobic parts. In this respect, this sentence is misleading.**

CAMS provides the concentration of the hydrophilic and hydrophobic components of BC. The total BC must be calculated as the sum of the two components. To avoid misunderstanding, the sentence was modified as:

... *"While $NO_2$ is a direct CAMS product, the ARCTEx BC mass concentration was calculated as the sum of the hydrophilic and hydrophobic components of BC provided by CAMS (Li et al., 2024). Hence, in our analysis, BC was treated as a single component without distinguishing the hydrophilic and hydrophobic fractions."* ...

**274: A plot of the rBC mass distribution and the fit would be useful.**

The use of the fitting curve to estimate the rBC mass concentration below and above the SP2 size limits is a common procedure for the SP2 data treatment. However, the reviewer comment rises an interesting point. The mass concentration outside of the size detection range is relatively minor compared to ambient observations, due to the narrow size distribution produced by the cast burner. Overall, the fitting allowed recovering the lost rBC mass below the size detection limit in the first 6 hours of each experiment, when the rBC size distribution peaked near the lower size detection limit. Afterwards, coagulation processes pushed the size distribution within the lower and upper detection limit of the SP2, removing the need for correction. The following statements were added in the main text, while a figure was included in the supplementary.

... *"The missing mass was attributed to the lower detection limit of the SP2 and the relatively small size of the injected particles (Figure **Error! Reference source not found.** in Section **Error! Reference source not found.** of the supplementary material). During the first two hours after injection, the undetected mass was estimated to range between 2% and 5%. However, as coagulation-driven growth progressed, the rBC particles fully shifted into the SP2 detection range within approximately six hours, effectively eliminating the initial under-detection issue."* ...

[Figure]

*Figure S 1 Example of mass size distribution and lognormal fit to estimate the mass concentration of rBC particles below the detection limit of the SP2. Dots: mass size distribution as measured by the SP2, organized in 50 bins*

*covering the 51-370 nm diameter range. Green line: lognormal curve fitted to the measured mass size distribution. Yellow area: estimated mass below the 51 nm, used to correct the rBC mass concentration of the SP2. The example represents the first hours of summer low altitude experiment.*

**302-303: "Since the concentrations of sulfate, ammonium, and chloride consistently remained below the detection limit" >> yes, because you did not inject.**

This makes sense, the sentence was modified as follows:

*… "Since sulfate, ammonium, and chloride were not introduced into the AIDA chamber, their concentration consistently remained below the detection limit. Hence, the coating was assumed to consist solely of nitrate and organic components."*

**304: Source of organic components? The burner only? Why is it representative of forest fires in Siberia? Or for other sources of Arctic BC?**

The answer of this comment was split in two parts:

- Answer to "Source of organic components? The burner only?": There were two sources of organic components inside AIDA. The first is primary organic carbon, produced during incomplete combustion and embedded in the soot matrix alongside elemental carbon. The mini-CAST burner was operated to minimize this fraction, which nonetheless remained present in lesser amounts. In our experiments, the organic carbon content at injection time ranged between 10% and 30%. The second source is secondary organic aerosol (SOA), formed through oxidation of volatile organic compounds (VOCs) co-emitted with soot during combustion. While the primary organic fraction was comparable across all experiments, SOA concentrations varied significantly depending on the conditions within AIDA. Details on the primary organic content of fresh soot are provided in Section C of the Supplementary Material, while SOA formation is discussed in Section 3.4 of the Results. To clarify the sources of organic material in AIDA, the text in Section 2.4.3 has been revised as follows:

  *… "The soot produced by the mini-CAST burner contains minimal organic carbon (Moore et al., 2014), but volatile organic compounds emitted as combustion by-products(Mamakos et al., 2013; Daoudi et al., 2023) may oxidize to form organic coatings (Lim et al., 2019). As a result, organic aerosol was observed during all experiments. In view of these considerations the coating was assumed to consist solely of nitrate and organic components." …*

- Answer to "Why is it representative of forest fires in Siberia? Or for other sources of Arctic BC?": We did not aim to reproduce forest fires emission, as described in the answer to a previous comment.

**228-229: Eq2 gives $d_f$ as the slope of the regression if you plot log(m) against log(k D^$d_f$) . In a log-log representation the slope has different meaning. This way the sentence and Fig S1c are ambiguous.**

The reviewer is right, the log-log was used as a representation in Figure S1, while the $d_f$ was calculated for each DMA-APM scan using a power law fit as actually described by Equation 2. The text was modified to clarify the message:

*… "Following Park et al. (2004), the relationship between particle mass and its diameter is described as a power law, where the exponent represents the fractal dimension ($d_f$) and $k_f$ a fit constant." …*

**353: Equation 4 gives the average material density of a particle with two components. The average material density is the same as the density of the particle if a solid core-shell structure is considered. Here this model cannot be applied since the core is a spongy structure (fractal-like particle), so the "coating" not only covers the surface but also fills the holes and cavities of the core particle. Thus, the coating increases the density through the compactness of the particle even if it has the same material density as the core particle. It would therefore be better to use the term "average material density".**

We agree with the consideration of the reviewer's comment, which made us realize that the effective to particle density ratio discussed in the result section was never properly introduced. Thus, we modified the text in Section 2.4.5:

… *"We note that this approach assumes a simplified dual-component system with solid and void-free core-shell structure and does not capture the morphology change induced by coating formation. Hence, the calculated $\rho_p$ should be interpreted as an average material density. The ratio $\rho_p/\rho_e$ was used as an indicator of particle compaction. Values near unity suggest a compact particle with internally mixed structure and minimal voids, while lower values indicate a fractal morphology, where internal voids reduce the effective density relative to the material density."* …

**380: Eq8 is only valid for spherical solid core-shell structures (for the reason mentioned above). The actual coating thickness can be obtained from the SP2 measurement. It would be interesting to compare the measured and calculated coating thicknesses, which may give an indication of the core particle compactness.**

The reviewer is correct. One of the technical objectives of this study was to compare the coating thickness derived from DMA–APM–SP2 measurements (as reported in the manuscript) with that obtained using the leading edge only fit (LEO-fit) method applied to SP2 data. LEO-fit is typically applied within a defined range of rBC core diameters. The specific SP2 instrument used during ARCTEx was previously employed to retrieve coating thicknesses of Arctic rBC particles in the 200–260 nm range (Zanatta et al., 2018). During ARCTEx, rBC core growth occurred via coagulation in all experiments; however, the majority of the rBC number size distribution remained below 200 nm. The fraction of rBC cores exceeding 200 nm was lower than 1% by the end of each experiment, making counting statistics the first limiting factor. A second limitation was related to the encapsulation of rBC cores. While internal mixing led to increased mobility diameters, the optical diameters remained below or close to the detection threshold of the SP2's scattering and position-sensitive detectors. We attempted to apply the LEO-fit method to the WL experiment, when the highest degree of internal mixing was observed, but we were unable to extract a sufficiently clean scattering signal from particles with rBC cores in the 200–250 nm range to apply LEO-fit reliably. Given the high uncertainty associated with coating thickness estimates derived from LEO-fit, we initially hoped that our results could help clarify the robustness of the method. However, we concluded that dedicated experiments are needed to specifically evaluate this technique, based on size selected rBC cores and thicker coatings.

These technical limitations were briefly mentioned at the end of Section 2.4.6 "Volume equivalent diameter and coating thickness", and we have now expanded that discussion to provide more details.

… *"It must be noted that the SP2 is also capable of estimating the coating thickness of rBC-containing particles (Laborde et al., 2012) by applying the leading edge only fit (LEO-fit) as proposed by Gao et al. (2007). To apply the method with this specific SP2 model, it is essential to obtain signals from the scattering and position-sensitive detectors with a high signal-to-noise ratio for particles having an rBC-core diameter in the 200-260 nm range (Zanatta et al., 2018). On one side, the number fraction of rBC cores exceeding 200 nm in diameter remained below 1% throughout our experiments, limiting counting statistics. More importantly, even if internal mixing led to increased mobility diameters during the SL and WL experiments, the optical diameters of the rBC-containing particles remained below or close to the detection threshold of the SP2's scattering and position-sensitive detectors. Hence, although the approach proposed here involved the use of several instruments, it represented*

*the only technical solution to determine the coating thickness for small and thinly coated BC-containing particles." ...*

Additionally, a cross-reference to Section 2.4.6 has been added to the SP2 description in Section 2.4.2 "Concentration and size distribution of total and refractory black carbon aerosol".

*... "Due to suboptimal combination of low signal-to-noise ratio of the scattering and position sensitive detectors, and the small optical size of the particles present in the AIDA chamber, it was not possible to quantify the coating thickness with SP2 measurements as proposed by Gao et al. (2007). More details are given in Section Error! Reference source not found.." ...*

**Figure 3: The discrepancy between CASM and AIDA RH and NO2/BC values should be discussed. Especially for WL and WH scenarios. How do those differences affect the representativeness of the simulation?**

Since this point was raised by multiple reviewers we addressed it in detail. Besides representing a major air transport pattern, the region of interest was chosen also because it shows a limited longitudinal variability in surface temperature, proving relatively stable atmospheric conditions. This was made clear at the end of Section 2.1.1 "Region of interest":

> *... "Although other efficient transport patters exist, this specific continental area is characterized by a reduced temperature variability (Przybylak, 2016), ensuring more homogeneous atmospheric conditions compared to the Atlantic and pacific transport pathways.*

e. To provide a better context to the reanalysis results, a short sub chapter was added in Section 3.1 "Northward transport conditions:

> *... "These temperature and humidity profiles are consistent with general Arctic conditions, which typically feature stronger latitudinal cooling and more humid conditions near the surface compared to higher altitudes (Przybylak, 2016). The longer atmospheric lifetime of $NO_2$ during winter (Levy II et al., 1999) may be responsible for the high $NO_2/BC$ in winter then in summer at both low and high altitude scenarios. Although elevated $NO_2$ relative to BC concentration is a prerequisite for nitrate coating formation, the production pathways in the Arctic are strongly influenced by extreme environmental factors such as low temperatures (Alexander et al.; 2020) and limited sunlight (Schaap et al., 2004), which affect both the efficiency and timing of nitrate formation. As reported by the recent AMAP (2021) report, the limited horizontal and vertical coverage of $NO_2$ measurements does not allow for further comparison with ambient data." ...*

f. Comparability of simulated and measured conditions. Considering that the estimated and observed temperatures were in close agreement we focused on relative humidity and $NO_2/BC$ ratio. The mismatch in RH may have affected gas-phase aqueous-phase chemistry processes controlling the formation of coating-precursors and coating material. The most affected scenario would be WL, when RH values were underestimated by approximately 10-15% during the entire duration of the experiment. The effect of $NO_2/BC$ mismatch would be, however, dominant on the coating formation potential. In this regard, we might have introduced contrasting bias in the low altitude scenarios. We have added a short paragraph at the end Section 3.1 "Northward transport conditions" to explicitly discuss the implications of these discrepancies. The modified text in section 3.1 now reads:

> *... "The mismatch in RH may have affected gas-phase and aqueous-phase chemistry processes influencing the formation of coating-precursors and coating material. As shown by the high values of the standard deviation of the $NO_2/BC$ ratio measured in the AIDA chamber, the control of NO2 concentration proved to be the most complicated. This technical difficulty led to non-systematic bias during the various experiments. As an example, we introduced contrasting bias in the low altitude scenarios. While the high $NO_2/BC$ in SL might promote a higher degree of internal mixing in AIDA compared to CAMS, the depleted level of NO2*

*might have hindered coating formation and internal mixing in WL. In summary, atmospheric conditions extracted from ERA-5 and CAMS revealed complex and heterogeneous transport conditions, which were well reproduced, day-by-day, in the AIDA chamber. Nonetheless, discrepancies between reanalysis and measured NO2 levels might accelerate or hinder coating formation in non-systematic way across the experiments."* ...

g. Figure 3 was modified to accommodate the comments done by different referees. The modifications include: i) shading representing the standard deviation of both reanalysis and measurements; ii) different colouring to improve readability; iii) adjustment of axis labels and legend.

[Figure]

*Figure 3 Left axis: Latitudinal profiles of temperature and relative humidity extracted from ERA-5 and of NO2/BC mass ratio extracted from CAMS in the region of interest (40-90°N and 60-140°E). Mean and standard deviation (SD) calculated for equidistant latitude bands 10° wide. Right axis: temporal variability of temperature, relative humidity and NO2/BC ratio measured in the AIDA chamber. Mean and standard deviation (SD) calculated over 24 hours.*

**468: Any explanation why the size distributions of the low and high scenarios do differ? And SL and WL? Could the chamber temperature/pressure affect the particle properties after injection?**

One parameter for explaining the observed differences in fresh rBC particle size distributions across scenarios could be the duration of injection. During the preparation phase of ARCTEx, we conducted a series of tests in a smaller vessel (aerosol preparation and characterization chamber, APC; Wagner et al., 2024) under ambient conditions and found a high degree of reproducibility in particle diameter during short injections (<1 minute), while longer injection times promoted rapid coagulation and thus larger particles. Considering the long injection time during ARCTEx scenarios (40-50 minutes), minor differences between SH and WH scenarios could be attributed to small variations (minutes) in injection duration. More details on injection duration were added in the method Section 2.4.1 and former Section B of the supplementary material.

Also chamber temperature could play a dominant role. Lower temperatures may enhance the immediate condensation of semi-volatile organic compounds in the first seconds after injection. This rapid condensation could compact the particle structure, potentially leading to a decrease in measured diameter (as seen in Figure S1a). This mechanism may also explain the observed increase in effective density from the warmer experiments

(WS–WL, −9 to −25 °C) to the colder ones (SL–WH, −35 to −52 °C). That said, we do not feel confident elaborating on these mechanisms in the main text, as coagulation processes may counteract condensation-driven shrinkage. A brief note has been added to former Section B of the supplementary material:

*... "The low density and diameter values observed in SH and WH suggest that the lower injection temperatures (−35 to −52 °C) compared to the warmer temperatures in SL and WL (25 to −9 °C)may have promoted early condensation of semi-volatile compounds on the soot cores, leading to compaction of the originally ramified structure. However, the fractal dimension and rBC mass fraction did not fully confirm this hypothesis."* ...

**Figure 4 caption: what do you mean "Concentrations adjusted to the ambient conditions inside the AIDA chamber."**

We have now clarified that all concentrations are adjusted to the actual in situ conditions (temperature and pressure) inside the AIDA chamber at the time of measurement, meaning that they are not normalized to standard conditions. This better reflects the physical conditions under which the aerosol processes occurred. The caption of Figure 4 was modified together with the final statement of Section 2.4.2.

- Modified caption: ... *"Concentrations normalized to the gas temperature and pressure of the AIDA chamber at the time of measurement."* ...
- Modified text: ... *"All number and mass concentrations reported hereafter refer to the actual temperature and pressure conditions inside the AIDA chamber (i.e., not normalized to standard conditions) at the time of the measurement."* ...

**514: Figure 6 is not discussed in the text.  521: Where is the FM$_{org}$ result presented? A relevant figure should be added. 548: Same for FM$_{NO3}$. 553: same**

We thank the reviewer for these observations. Although Figure 6 does show the evolution of the mass fractions of organic and nitrate coatings (FM$_{org}$ and FM$_{nit}$), it was mentioned only briefly at the beginning of Section 3.4 ("Evolution of BC morphology and mixing state during aging"). This limited referencing, as in the case of Figure 5, aimed to keep the text concise and avoid excessive cross-referencing. However, we recognize that this may have unintentionally made it harder for the reader to connect specific parts of the text with the relevant figures. In response, we revised Section 3.4.1, Section 3.4.2 and Section 3.4.3 to include more frequent references to Figures 5 and 6.

**571-572: Do you have any explanation for this? Temperature effect? Slower chemistry?**

We hypothesize that temperature plays the dominant role in preventing the formation of thicker coatings. However, several reaction pathways involving ozone, nitrogen oxides and volatile organic compounds might take place and compete.

Considering nitrate coatings formation during day time, the photolysis of ozone is the major production pathway of OH radicals in the atmosphere (Finlayson-Pitts and Pitts, 1986), which in turn oxidizes $NO_2$ to form $HNO_3$

- $O_3 + h\nu \ \rightarrow O(1D) + O_2 \quad (\lambda \leq 310 \ nm)$
- $O(1D) + H_2O \ \rightarrow 2OH$
- $NO_2 + OH \ \rightarrow HNO_3$

During nighttime, the lifetime of $NO_3$ (formed from the oxidation of $NO_2$ in presence of ozone) is long enough to trigger the formation of $HNO_3$ via

- Reaction with volatile organic carbon (Ng et al., 2017):
  - $NO_3\,(g) + DMS/CH(g) \longrightarrow HNO_3\,(g)$
- Heterogeneous hydrolysis of $N_2O_5$ (Alexander et al., 2020):
  - $NO_3\,(g) + NO_2(g) \rightleftarrows N_2O_5\,(g)$
  - $N_2O_5\,(g) + H_2O\,(l) \rightarrow HNO_3\,(l)$

Gas-phase loss of HNO$_3$ must be considered as well (Dulitz et al., 2018):

- $HNO_3\,(g) + OH(g) \rightleftarrows NO_3\,(g) + H_2O$
- $HNO_3\,(g) + h\nu \rightarrow OH + NO_2$

According to literature, the photolysis rate of ozone slightly decreases with temperature (Matsumi et al., 2002), potentially decreasing the OH radical formation and nitric acid formation during day time. Cold temperature would however prolong the lifetime of $N_2O_5$, promoting the hydrolysis reaction pathway (Alexander et al., 2020), but also increase the reconversion of nitric acid to nitrogen oxides (Dulitz et al., 2018).

Secondary organic aerosol (SOA) coatings may be produced from volatile organic compounds via several pathways (Srivastava et al., 2022). Although cast burners were proven to emit volatile organic compounds, polycyclic aromatic hydrocarbons and alkanes (Mason et al., 2020), their specific reactivity and SOA formation yields were never reported. However, the review of Srivastava et al. (2022) indicated a positive effect of RH on SOA yields, while temperature had a negative effect or little influence on SOA formation due to enhanced partitioning into the condensed phase at lower temperature. At subzero temperature, relevant for the high troposphere, the lifetime of biogenic VOCs in presence of ozone is highly enhanced (Tillmann et al., 2009). Overall, considering the formation of secondary organic aerosol coatings, anthropogenic organic emissions are still poorly constrained (Jathar et al., 2014)

In view of these details, at high temperature and prolonged light conditions (SL), O$_3$ was consumed quickly at a maximum rate of 10 ppm per hour immediately after injection. However, the hourly decrease of NO$_2$ remained almost constant, leading to an almost negligible formation of nitrate. In this scenario with warm conditions and low NO$_2$/BC ratio, SOA formation was clearly dominant. In WL, during irradiation time, the hourly consumption of O$_3$ was similar to SL, but was followed by a marked decrease of NO$_2$. This indicates an efficient photolysis of ozone, OH radical formation and conversion to nitric acid and nitrate. In contrast, nighttime formation and deposition of nitrate presumably play a secondary contribution. These two experiments might confirm that the reactivity of VOCs with ozone is extremely efficient above 0 °C and competes with the oxidation of nitrate during day and nighttime. In summarized form, these considerations were provided in the main text in Section 3.4.1 and Section 3.4.2.

During the high-altitude scenarios, we observed an almost constant hourly depletion of NO$_2$ and O$_3$, largely independent of light conditions. Notably, their depletion rates closely followed that of CO$_2$ (used as a dilution tracer), suggesting limited chemical reactivity at temperatures below −30 °C. This supports the hypothesis that temperature plays a dominant role in suppressing both daytime and nighttime chemistry under these conditions. It must also be considered that the water vapour pressure was much lower in the coldest experiment, potentially limiting OH formation and hydrolysis of $N_2O_5$. However, our experimental setup did not permit the identification and quantification of all relevant organic and inorganic precursors, nor the absolute determination of the coating mass. As a result, the temperature effect on the coating formation "yield" remains unquantifiable. Given these limitations, a detailed discussion of specific reaction pathways would be speculative. For this reason, and in line with the scope of this study, we did not elaborate further on this in Sections 3.4.1 and 3.4.2. Nonetheless, we briefly raised our hypothesis and the current limitations at the end of Section 3.4.3 ("Slower aging during high-altitude transport"). We are open to include the extended discussion provided here in the supplementary material, in case considered necessary by the reviewers and editor.

*... "This aging reduction, was likely linked to low temperatures, which may slightly reduce the yield of ozone photolysis (Matsumi et al., 2002), shorten the lifetime of HNO₃ (Dulitz et al., 2018), and minimize the oxidation of volatile organic compounds to secondary organic aerosol coatings (Saathoff et al., 2009; Tillmann et al., 2009). This hypothesis is reinforced by the near-constant depletion rates of NO₂ and O₃, irrespective of light conditions,*

*suggesting limited photochemical and nocturnal reactivity at temperatures below -30 °C. However, due to the lack of comprehensive gas-phase (nitric acid, OH radical, volatile organic compounds) and particle-phase (absolute quantification of nitrate, organic and organo-nitrate compounds) chemical speciation, we cannot quantify the coating yields or attribute them to specific reaction pathway."* ...

**Figure 5: Note which are the direct measurement results, and which are calculated ones (e.g. particle density)?**

All the variables presented in Figure 5 are calculated combining multiple instruments, and they are fully described in the technical section and summarized in the annex table. To avoid repetitions, we chose not to modify the caption of Figure 5.

**Figure 5: What are the gaps between the points?**

The properties shown in Figure 5 are derived from DMA-APM and DMA-APM-SP2. These measurements were not performed continuously but, usually, before and after the virtual midnight (change of conditions). This is now specified at the end of the introduction of Section 3.4 "Evolution of BC morphology and mixing state during aging" :

... *"Excluding $FM_{nit}$ and $FM_{org}$, the other properties described in this section were calculated from the DMA-APM and DMA-APM-SP2 scans. These scans were performed before and after the virtual midnight, resulting in a low temporal resolution."* ...

**625-626: This finding should be more elaborated. What is the exact relationship you found between coagulation growth and coating?**

The slow ageing observed during the prolonged experiments in AIDA allowed noticing several side processes. Among other observations, we noted an abrupt flattening of the exponential growth of rBC cores (Figure S3a) at t24 during WL. At t24, rBC particles increased density, and fractal dimension approached spherical limits (Figure 5). Hence, we hypothesized that increase of fractal dimension, corresponding to a decrease in the active particle cross section, limited the collision efficiency reducing the coagulation rate. To the best of our knowledge, this process was simulated with numerical models but never really observed with in-situ measurements.

- The statement in the main text was simplified as:

  ... *"The concurrent increase in fractal dimension may have reduced active surface area, limiting collisions and slowing coagulation. Despite prior studies (e.g. Schnaiter et al., 2003; Maricq, 2007), coagulation during the transition from external to internal mixing remains poorly characterized."* ...

- More details are now given in Section E of the Supplementary material:

  ... *"In WL, the diameter growth turned from an exponential function to a linear function in WL after the second aging step (t24), when fractal dimension and coating mass fraction increased (50°N; **Error! Reference source not found.**). Particle restructuring at the onset of coating may have reduced the active surface area, thereby limiting the collision frequency and coagulation rate. This hypothesis aligns with numerical simulations of Naumann (2003), suggesting that coating formation may reduce coagulation rates near the sources or in polluted regions. Although particle coagulation has been extensively studied in other settings (Schnaiter et al., 2003; Maricq, 2007), our finding raises the need of targeted experiments to quantify the change of coagulation rate of soot particles in the transition regime from external to internal mixing in ambient like conditions."* ...

**661: More explanation and/or reference is needed for Hill equation.**

The Hill equation is a sigmoidal curve mostly used in biology/pharmacology to describe saturable processes or temporal appearance of events (Weiss, 1997; Goutelle et al., 2008). Hill equation was chosen to represent a process with well-defined minimum (bare rBC) and maximum (fully coated rBC) $Rm_{coat}$ asymptotes. The upper limit was considered asymptotic due to the dominant role of coating mass over an almost negligible mass of rBC core, within the ARCTEx experimental time. Several tests performed with more generalized Sigmoid functions indicated that the Hill equation was able to capture the sudden increase of $Rm_{Coat}$, showing smaller residuals, at low elapsed time. The text was modified as:

*... "Next, we derived the aging timescale necessary to reach the critical coating mass for each scenario. We applied a Hill equation (Weiss, 1997; Goutelle et al., 2008) to model the non-linear relationship between $Rm_{coat}$ and BC age, forcing the upper (maximum $Rm_{coat}$ observed during WL) and lower (initial $Rm_{coat}$ from each scenario) boundaries along with a 95% confidence interval (**Error! Reference source not found.**a, dashed lines). Compared to a more generic sigmoid equation, the Hill function allowed capturing the sudden increase of $Rm_{coat}$, in the early phase of the ageing." ...*

**Figure S1: Mention the instrument/technique used to obtain the results. For example, plot a) is an SMPS result I guess? Plot c) APM vs. SMPS and so on.**

The caption of the figure now includes the name of the instruments.

**Figure S1a: Any explanation why the size distributions of the low and high scenarios do differ? And SL and WL? Could the chamber temperature/pressure affect the emission after injection?**

This comment was addressed above.

**Figure S3: only 3 curves are visible in panel a). WH is missing, I guess.**

The reviewer is correct - panel a of Figure S3 appears to show only three visible curves because SH and WH are almost perfectly overlapping. While this overlap may reduce the immediate readability of the figure, it also highlights the excellent repeatability of our experiments under conditions where no significant internal mixing occurred. We increase the resolution of the image to facilitate the reading.

**References**

[revised manuscript text omitted]

**Answers of the authors to Reviewer#3**

While the reviewer's comments are given in **black bold**, our answers are given below in blue letters.
Additionally, we added the changes made in the revised manuscript in *blue italic letters*.

The manuscript titled "AIDA Arctic transport experiment (part 1): simulation of northward transport and aging effect on fundamental black carbon properties" presents a well-structured and insightful experimental investigation into black carbon (BC) aging under conditions representative of Arctic transport. To accurately quantify and understand the impact of atmospheric aging on BC properties and radiative forcing, the ARCTEx project simulated BC aging under quasi-realistic Arctic conditions in the AIDA. Informed by reanalysis data, four distinct scenarios were developed to capture seasonal and altitudinal variability during Arctic transport. The use of the AIDA chamber to emulate these variations is methodologically sound and lends credibility to the experimental design. The results on coating composition, morphological evolution, and aging timescales provide valuable empirical constraints for improving the representation of BC aging in atmospheric models.

We would like to thank the referees for their detailed and constructive comments, which helped us to improve our manuscript. Here we provide some major considerations. Several reviewers noted an insufficient discussion on the ambient representativity of transport conditions, soot generation and coating species. This was now addressed in more detail in both the Methods and Results sections, representing the major modification to the text. Reviewers also identified inconsistencies in the use of acronyms and abbreviations. These have been reviewed and corrected throughout the manuscript text and figures. The reviewer's specific comments are addressed as it follows.

**Specific comments**

**While the dominance of nitrate and organic coatings is clearly demonstrated, the absence of sulphate or ammonium in coatings (Section 2.4.3) should be discussed. Clarifying this would help readers assess the generalizability of the findings.**

This point was made by other reviewers as well, so the text was modified in several sections to better explain this point. We intentionally excluded sulfate, ammonia, and chlorine to focus on nitrate and organic ageing processes. Even with a simplified chemical system, ARCTEx showed that ageing under changing environmental conditions is still complex. This evolving-ageing setup raised additional questions, pointing to the need for further, focused experiments to better understand specific processes like organic and nitrate competitive oxidation, temperature-driven coating formation and inhibition of coagulation by coating. More detailed explanations are given as follows.

- The aim was to investigate coating with nitrate which have been often disregarded in the past. Hence, in the introduction we now describe the relevance of studying nitrate species and their interaction with BC: ... *"BC variability in the Arctic was often associated with co-emitted sulfate aerosol from anthropogenic sources (e.g. Massling et al., 2015) and organic aerosol from biomass burning events (e.g. Moschos et al., 2022), while its correlation with nitrate was mostly ignored (AMAP, 2021). Similarly, chamber studies focused on the evolution of BC properties as function of internal mixing with sulfate (e.g. Möhler et al., 2005; Khalizov et al., 2009; Henning et al., 2012) and organics (e.g. Lefevre, 2019; Wittbom et*

*al., 2014). As a consequence, the impact of BC-nitrate internal mixing on fundamental and climate relevant properties remained poorly assessed (Yuan et al., 2020). Internal mixing of BC with nitrate species becomes particularly important in the Arctic region, where, nitrate aerosol concentration have been increasing since the '80s despite an overall reduction of nitrogen oxides emissions (AMAP, 2021). The same report underlined how few studies focused on nitrate aerosol in the Arctic, introducing a knowledge gap on the sources of its precursors, its formation mechanisms and its interaction with other atmospheric species such as BC."* ...

- The volatile organic aerosol, which may oxidize to secondary organic aerosol and condense on BC represented a non-controlled source of coating precursors, which is representative of the mini-CAST burner. This point was explicitly specified in Section 2.1.4 "Atmospheric composition":

  *... "It must be noted that volatile organic compounds which are a by-product of combustion, were simultaneously emitted with BC and injected in the AIDA chamber without active control. As a result, the organic aerosol content in AIDA reflects the specific emissions of the burner and not ambient-like conditions. Therefore, although the experiments primarily targeted the evolution of BC mixing with nitrate coatings, the presence of organic vapours may interact or compete with $NO_2$ during condensation and coating formation, introducing additional complexity to the aging dynamics."* ...

- We did not observe sulphate or ammonia coatings because they were neither injected directly, nor co-emitted with soot by the mini-CAST burner. This is now stated properly in Section 2.4.3 "Chemical characterization of non-refractory aerosol particles":

  *... "Since sulfate, ammonium, and chloride were not introduced into the AIDA chamber, directly or indirectly as a byproduct of combustion,"* ...

**Section 3.5: Briefly justify the use of the Hill equation for modelling aging timescales.**

As also requested by Reviewer 2, more details on the Hill equations were added in Section 3.5 "Aging timescales":

*... "Next, we derived the aging timescale necessary to reach the critical coating mass for each scenario. We applied a Hill equation (Weiss, 1997; Goutelle et al., 2008) to model the non-linear relationship between $Rm_{coat}$ and BC age, forcing the upper (maximum $Rm_{coat}$ observed during WL) and lower (initial $Rm_{coat}$ from each scenario) boundaries along with a 95% confidence interval (**Error! Reference source not found.**a, dashed lines). Compared to a more generic sigmoid equation, the Hill function allowed capturing the sudden increase of $Rm_{coat}$, in the early phase of the ageing."* ...

**The manuscript could be further strengthened by discussing implications for global or regional climate models, particularly in light of recent developments in BC aging parameterizations (e.g., https://doi.org/10.5194/acp-25-2613-2025 and DOI:10.1029/2024JD041135).**

- We really appreciated this comment, because it reinforces the novelty of our findings which fits in a recent set of publications raising an interest towards the importance of ageing timescale of BC. These works include Chen et al. (2024), Fierce et al. (2025) and Jin et al. (2025). Modifications were done in Section 3.5
- To reflect the finding of the Section, its title was changed from "Ageing timescale" to "Varying ageing timescales as function of transport pathway".
- A brief introduction to summarize how ageing is treated in models is now given at the beginning of the section:

  *... "Global models parametrize this conversion in several ways. The most simplified approach in bulk aerosol models is to consider a fixed ageing timescale (Koch et al., 2009). Other aging schemes, used in modal aerosol modules, include a coating thickness threshold made up of a variable number of mono-layers (Liu et al., 2016). Considering the large discrepancies between observations and simulations, more detailed treatment of size, morphology and mixing is implemented in the modules of global models (e.g. (Matsui, 2016; Chen et al., 2024; Jin et al., 2025))."* ...

- We describe our findings in the context of modelling results at the end of the section:

*... "Our experimental results reinforce the recent findings of Fierce et al. (2025), who highlighted the inadequacy of fixed ageing timescales in models. Their work confirms that ageing rates are regionally and seasonally dependent, as observed in the ARCTEx scenarios, significantly affecting simulated BC concentrations, particularly in the Arctic. Moreover, the altitudinal and seasonal ageing patterns shown in* **Error! Reference source not found.** *reflect ambient variability and lead to heterogeneous impacts on the hygroscopic and optical properties of BC (Jin et al., 2025)." …*

**Figure 3: Adding error bars or uncertainty shading would help visualize variability.**

In the revised version of the manuscript, we have expanded the discussion regarding simulated vs. measured conditions. Overall, focussing on the NO2/BC ratio, we might have introduced some bias in the AIDA simulations compared to the ambient variability, especially in the low altitude scenarios. This is now discussed in Section 3.1 "Northward transport conditions". Following the request of multiple reviewers, we also modified Figure 3 adding the standard deviation of the reanalysis data and AIDA measurements. The modifications include: i) shading representing the standard deviation of both reanalysis and measurements; ii) different colouring to improve readability; iii) adjustment of axis labels and legend.

[Figure]

*Figure 3 Left axis: Latitudinal profiles of temperature and relative humidity extracted from ERA-5 and of NO₂/BC mass ratio extracted from CAMS in the region of interest (40-90°N and 60-140°E). Mean and standard deviation (SD) calculated for equidistant latitude bands 10° wide. Right axis: temporal variability of temperature, relative humidity and NO₂/BC ratio measured in the AIDA chamber. Mean and standard deviation (SD) calculated over 24 hours.*

**Line 599 and 619: The unit "kg m³" should be corrected to "kg m⁻³".**
**Figure 8: The variable label "Rm_Coat" should be revised to "Rm_coat" to ensure consistency.**
**Table A1: Please verify that the list of abbreviations is complete. For example, "Rm_coat" appears in the manuscript but is missing from the table.**
**Ensure consistent use of notation, like kg m⁻³ in the manuscript and "Kg m⁻³" in the Table A1.**

We thank the reviewer for these comments. We verified the consistency of abbreviations and symbols in the manuscript and supplementary material and found many discrepancies. We now harmonized all abbreviations and symbols included in the text and figures according to the table in appendix.

**References**

[revised manuscript text omitted]